

# Kinetics of quantum reaction-diffusion systems

Federico Gerbino[1*], Igor Lesanovsky[2,3] and Gabriele Perfetto[2]

**1** Laboratoire de Physique Théorique et Modèles Statistiques,
Université Paris-Saclay, CNRS, 91405 Orsay, France
**2** Institut für Theoretische Physik, Eberhard Karls Universität Tübingen,
Auf der Morgenstelle 14, 72076 Tübingen, Germany
**3** School of Physics and Astronomy and Centre for the Mathematics and Theoretical Physics
of Quantum Non-Equilibrium Systems, The University of Nottingham,
Nottingham, NG7 2RD, United Kingdom

★ federico.gerbino@universite-paris-saclay.fr

## Abstract

We discuss many-body fermionic and bosonic systems subject to dissipative particle losses in arbitrary spatial dimensions $d$, within the Keldysh path-integral formulation of the quantum master equation. This open quantum dynamics represents a generalisation of classical reaction-diffusion dynamics to the quantum realm. We first show how initial conditions can be introduced in the Keldysh path integral via boundary terms. We then study binary annihilation reactions $A+A \to \emptyset$, for which we derive a Boltzmann-like kinetic equation. The ensuing algebraic decay in time for the particle density depends on the particle statistics. In order to model possible experimental implementations with cold atoms, for fermions in $d = 1$ we further discuss inhomogeneous cases involving the presence of a trapping potential. In this context, we quantify the irreversibility of the dynamics studying the time evolution of the system entropy for different quenches of the trapping potential. We find that the system entropy features algebraic decay for confining quenches, while it saturates in deconfined scenarios.

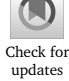

# 1  Introduction

The identification and classification of universal behaviour in critical equilibrium systems is one of the greatest achievements of statistical mechanics [1–4]. Thermal equilibrium is nonetheless a rather idealised situation in nature, since a huge variety of relevant processes take place far from equilibrium in driven or relaxational conditions. In that scenario, understanding the emergence of collective properties in many-body systems and, possibly, defining universality classes is a demanding task.

Reaction-Diffusion (RD) models stand out as prototypical cases of genuinely far-from-equilibrium systems where the calculation of critical exponents and the recognition of universal properties is made possible [5–10]. In classical discrete systems, diffusion is modelled by nearest-neighbour stochastic hopping, while reactions take place when two or more particles are located on the same lattice site. The ensuing stochastic dynamics is ruled by a classical master equation. Paradigmatic binary processes are, for instance, binary annihilation $A + A \rightarrow \emptyset$ and coagulation $A + A \rightarrow A$ reactions. These reactions provide a plethora of relaxational non-equilibrium dynamics. Therein reactions only deplete the system, and one has critical dynamics in the way the stationary state, devoid of particles, is eventually approached in time. In particular, one has that the particle density decays according to a power law with universal amplitude and exponent.

Two typical regimes can be identified in RD systems. The so-called *reaction-limited* regime of fast diffusive mixing and slow reactions [5, 7, 11–13], where the timescale of the dynamics is dominated by the reaction rate $\Gamma$. In particular, mean-field calculations provide the correct long-time asymptotics for the density $n(t)$ as a function of time $t$. For $A + A \rightarrow \emptyset$ and $A + A \rightarrow A$, this yields the decay law

$$n(t) \sim (\Gamma t)^{-1}. \tag{1}$$

In the opposite *diffusion-limited* regime [14–18], where diffusion and reactions compete at similar timescales, mean-field does not provide an accurate prediction of the long-time behaviour. Crucially, the development of Doi and Peliti's statistical path-integral approach [19, 20], made it possible to thoroughly access the diffusion-limited regime of classical RD dynamics via renormalisation group methods [8–10, 21–23]. Binary annihilation and coagulation are specifically shown to belong to the same universality class and they show algebraic decay with different exponent than mean field, Eq. (1), in one spatial dimension.

Determining the possible impact of quantum effects on emergent nonequilibrium behaviour is currently the aim of intense research. Deviations from classical predictions have indeed been identified for various kinetically constrained models [24–31] described by the Markovian quantum master equation in Lindblad form [32, 33]. Quantum RD dynamics is also formulated in terms of the Lindblad equation. Therein, stochastic diffusion is replaced by coherent hopping, while reactions are irreversible and modelled by dissipative quantum jump operators. However, investigations of RD dynamics in the quantum realm have only recently been conducted [34–48]. The ensuing many-body dynamics are indeed even harder to address than their classical counterparts due to the exponential scaling of the many-body Hilbert space with the system size. The assessment of large-scale RD non-equilibrium universal properties is therefore extremely challenging for quantum systems as it entails the simultaneous simulation of large sizes and long times. Recent numerical studies [49–53] therefore focused on one-dimensional small systems. From the latter results, however, it is hard to make unambiguous statements on the resulting behaviour in the thermodynamic limit.

Analytical results in the thermodynamic limit for one-dimensional quantum RD models have been recently obtained in Refs. [36, 38, 40, 42–44, 47, 48, 54–57] in the reaction-limited regime. For fermionic systems [38, 40, 42–44, 47, 54–56], it has been shown that the quantum reaction-limited RD dynamics yields algebraic decay for the particle density with different exponent than the mean-field one in Eq. (1). For noninteracting bosonic systems, instead, it has been shown [36, 57] that multi-body annihilation reactions $kA \rightarrow \emptyset$ ($k \geq 2$) lead to mean field decay for the particle density. In the case where the interacting Bose gas is considered [36], the asymptotic decay of the particle density is, instead, not known.

A common aspect of the works [36, 38, 40, 42–44, 54–57], both for fermions and for bosons, is that the underlying analysis is based on the the time-dependent Generalised Gibbs Ensemble (TGGE) ansatz [58–61] for the reaction-limited regime. Within the TGGE method, for weak reactions-dissipation, the state of the system in-between consecutive reactions is assumed to be a maximal entropy state consistent with all the conservation laws of the Hamiltonian. This state has the form of a GGE, see, e.g., the reviews [62, 63], which allows to derive exact dynamical equations for the occupation function in momentum space and hence the particle density. In Refs. [55, 56], instead, a different approach has been pursued. In these references, the Keldysh path integral representation, see, e.g., the reviews [64–67], of the quantum dynamics is considered. In Ref. [55], the one-dimensional Bose-Hubbard chain under strong two-body losses has been analysed via the Feynman-Vernon influence functional. The latter is obtained by performing a second-order cumulant expansion of the system-bath Keldysh action and integrating out the bath degrees of freedom. In the regime of dominant losses, the system maps to the reaction-limited dynamics of a Fermi gas, as in Refs. [34, 68]. In Ref. [56], instead, the Fermi gas in continuum space and in arbitrary spatial dimensions $d$ in the reaction-limited regime has been considered. Here, the dynamics for two-body losses is directly formulated in terms of the Lindblad-Keldysh action. The TGGE dynamical equation is then recovered by taking the Euler-hydrodynamic scaling limit [69, 70] of the diagrammatic expansion of the dissipative interaction vertices.

In this manuscript, we systematically present and further develop the results of Ref. [56]. Specifically, we detail the formulation of the quantum RD dynamics via Keldysh path integrals. We also show how to implement initial conditions within the Keldysh action. Initial conditions, namely, correspond to boundary terms where the fields are computed solely at the initial time $t = 0$. These boundary conditions add to the bulk Keldysh action. We benchmark the implementation of initial conditions against the exactly solvable case of a single-body decay $A \to \emptyset$. Therein, we show that the action composed of the bulk Keldysh action and the boundary-initial term correctly predicts an exponential decay for the particle density. Moving to more interesting, not exactly solvable, cases, we consider binary annihilation $A + A \to \emptyset$. The Keldysh path integral allows us to naturally derive kinetic equations. We accomplish this in generic spatial dimensions $d$ for both bosons and fermions. In both cases, we perform a diagrammatic expansion of the interaction vertices from the Keldysh partition function. In the Euler-scaling limit, $\Gamma \to 0$ with $\bar{\vec{x}} = \Gamma \vec{x}$ and $\bar{t} = \Gamma t$ fixed, of slow space-time variation, this expansion can be truncated at first order in space-time derivatives and it acquires the universal form of a kinetic Boltzmann equation. In $d = 1$, this approach is equivalent to the TGGE ansatz for the reaction-limited regime. For homogeneous bosonic systems, we obtain mean field decay for the particle density in all dimensions $d$, cf. Eq. (1). For homogeneous fermionic systems, instead, deviations in the decay exponent from mean field are observed in all dimensions $d$, as previously derived in [56]. In order to more closely describe cold-atomic experiments involving particle losses [71–76], where the quantum gas is inhomogeneous in space due to the presence of a trapping potential, we also consider inhomogeneous fermionic systems in one dimension with quenches of the trapping potential. For quenches from an anharmonic potential to a harmonic one, we find that the anharmonicity in the initial potential causes for the density a faster algebraic decay compared to both the homogeneous decay and the decay happening from an initial harmonic potential. Furthermore, we quantify the irreversibility of the dynamics due to dissipation by computing the dynamics of the system entropy. We observe that for quenches from an anharmonic to an harmonic potential the system entropy decays to zero algebraically. This decay is caused by the continuous loss of particles and the associated growth of the surrounding environment entropy. Interestingly, we find that the decay exponent of the system entropy coincides with the decay exponent of the particle density in homogeneous setups. This decay exponent is further observed not to depend on the anharmonicity of the initial potential (differently from the aforementioned decay of the density). We eventually consider trap-release quenches where the initial trapping potential is switched off. In this case, the quantum gas freely expands in space. We find that after an initial transient, reactions become scarcer and scarcer and the gas just expands in space according to the Euler equation. The system entropy therefore saturates in time as a consequence of the ballistic-reversible quantum transport of particles.

The remainder of the manuscript is organised as follows. In Sec. 2, we formulate the quantum RD dynamics in the terms of the Lindblad master equation. In Sec. 3, we recall the basic aspects of the Keldysh path integral needed for the understanding of our results. In Sec. 4, we show how the initial conditions of the dynamics can be inserted in the Keldysh path integral via boundary terms in addition to the bulk Keldysh action. In Sec. 5, we derive the Boltzmann equation in the Euler-scaling limit for the Bose and the Fermi gases in $d$ spatial dimensions subject to binary annihilation reactions $A + A \to \emptyset$. In Sec. 6, we eventually provide an application of the Boltzmann equation to the study of inhomogeneous fermionic systems with a trapping potential in $d = 1$. In Appendix A, we discuss some aspects of the Doi-Peliti path-integral formulation of classical RD systems, which are useful for comparison with the Keldysh quantum RD formulation. Additional details on the calculations are reported in Appendix B.

## 2 Quantum reaction-diffusion models

In this Section, we introduce the quantum RD models considered throughout the text. For the sake of simplicity, we start from a system on a lattice, discussed in Subsec. 2.1. In Subsec. 2.2, we obtain the associated continuum limit in space both for bosons and fermions.

### 2.1 The master equation formalism

We consider a $d$-dimensional lattice system of bosons or fermions. Each lattice site is identified by the array of indices $\mathbf{i} = (i_1, \ldots, i_\alpha, \ldots, i_d) \in \mathbb{Z}^d$ ($\alpha = 1, 2 \ldots d$) of a hyper-cubic lattice with $I^d$ sites, with $\mathbf{e}_\alpha$ the unit-vector pointing in the $\alpha$-th direction and $l$ the lattice spacing. The bosonic (fermionic) operators satisfy the canonical (anti)commutation relations

$$[a_{\mathbf{i}}, a_{\mathbf{j}}^\dagger]_\xi = \delta_{\mathbf{ij}}, \qquad [a_{\mathbf{i}}, a_{\mathbf{j}}]_\xi = [a_{\mathbf{i}}^\dagger, a_{\mathbf{j}}^\dagger]_\xi = 0, \tag{2}$$

with $[a, b]_\xi = ab - \xi ba$ and $\xi = +1$ for bosons, $\xi = -1$ for fermions. Throughout our discussion we will consider spinless fermions, which are frequently used in many-body physics. In $d = 1$, spinless fermions can be mapped to spins $1/2$ via Jordan-Wigner transformation [77]. The standard quantum mechanical normalisation of state vectors in Fock space is used:

$$a_{\mathbf{i}} |n_{\mathbf{i}}\rangle = \sqrt{n_{\mathbf{i}}} |n_{\mathbf{i}} - 1\rangle, \qquad a_{\mathbf{i}}^\dagger |n_{\mathbf{i}}\rangle = \sqrt{n_{\mathbf{i}} + 1} |n_{\mathbf{i}} + 1\rangle. \tag{3}$$

Clearly, fermions satisfy the additional constraint $n_i = 0, 1$ due to the Pauli exclusion principle. The dynamics of the quantum many-body density matrix $\rho$ is governed by a quantum master equation in Lindblad form [32, 33]:

$$\dot{\rho}(t) = \mathcal{L}[\rho(t)] = -\frac{i}{\hbar}[H, \rho(t)] + \mathcal{D}[\rho(t)]. \tag{4}$$

The Lindblad map is trace-preserving, namely, at any instant of the evolution one has $\mathrm{Tr}\rho = 1$ (provided the initial density matrix has also trace 1). The Hermitian Hamiltonian $H$ in Eq. (4) determines the unitary evolution: in our quantum RD settings, we consider the free hopping Hamiltonian

$$H = -\frac{J}{l^2} \sum_{\mathbf{i}} \sum_{\alpha=1}^d (a_{\mathbf{i}}^\dagger a_{\mathbf{i}+\mathbf{e}_\alpha} + a_{\mathbf{i}+\mathbf{e}_\alpha}^\dagger a_{\mathbf{i}}) + \sum_{\mathbf{i}} a_{\mathbf{i}}^\dagger V_{\mathbf{i}} a_{\mathbf{i}}, \tag{5}$$

where $J/l^2$ [units $\hbar$ time$^{-1}$] is the hopping rate. This parameterisation of the hopping rate is chosen so that in the continuum limit, discussed in Subsec. 3.1, $J$ has units [length$^2 \cdot$ time$^{-1}$]. The term $V_{\mathbf{i}}$ represents an external single-body potential. In Sec. 5, we will consider the potential $V_{\mathbf{i}}$ to vary on a macroscopic length scale $\ell \gg l$. This will allow us to derive the Boltzmann equation in the Euler-scaling limit. Eq. (5) provides a quantum generalisation of the stochastic hopping considered in classical RD models. It is, however, important to note that in classical RD, stochastic hopping on the lattice amounts to diffusive transport of particles in the continuum. In the quantum case, Eq. (5) gives ballistic coherent transport of particles. The Lindblad dissipator $\mathcal{D}$ encodes irreversible reaction processes, and it is usually written in terms of the quantum jump operators $L_{\mathbf{i},\alpha}$:

$$\mathcal{D}[\rho] = \sum_{\mathbf{i}} \sum_{\alpha=1}^d [L_{\mathbf{i},\alpha} \rho L_{\mathbf{i},\alpha}^\dagger - \frac{1}{2}\{L_{\mathbf{i},\alpha}^\dagger L_{\mathbf{i},\alpha}, \rho\}]. \tag{6}$$

We consider throughout the manuscript two different reactions. First, we study the case of one-body decay $A \rightarrow \emptyset$ at rate $\Gamma_d$ in Sec. 4 [units time$^{-1}$]:

$$L_{\mathbf{i},\alpha} = \frac{1}{d}\sqrt{\Gamma_d}\, a_{\mathbf{i}}. \tag{7}$$

Second, we consider pair annihilation $A + A \rightarrow \emptyset$ at rate $\Gamma$ in Secs. 5 and 6. The form of the associated jump operators depends on the quantum statistics of the particles. In the bosonic case, we consider the annihilation operator $L_{\mathbf{i}}^{\mathrm{B,ann}}$ given by:

$$L_{\mathbf{i},\alpha}^{\mathrm{B,ann}} = \frac{1}{d}\sqrt{\frac{\Gamma}{l^d}}a_{\mathbf{i}}^2, \tag{8}$$

with $\sqrt{\Gamma/l^d}$ the annihilation reaction rate. In the fermionic case, the exclusion principle implies that, within the quantum RD processes scenario, two identical fermions cannot overlap, and reactions must occur between nearest neighbours. Accordingly, we define the following fermionic annihilation operator

$$L_{\mathbf{i},\alpha}^{\mathrm{F,ann}} = \sqrt{\frac{\Gamma}{l^{d+2}}}a_{\mathbf{i}}a_{\mathbf{i}+\mathbf{e}_\alpha}. \tag{9}$$

In Eq. (8), the constant $\Gamma$ has units $[\mathrm{length}^d \cdot \mathrm{time}^{-1}]$. In Eq. (9), instead, $\Gamma$ has units $[\mathrm{length}^{d+2} \cdot \mathrm{time}^{-1}]$.

## 2.2 The continuum limit

The continuum-space limit is identified by considering infinite sites $I^d \rightarrow \infty$, vanishing lattice spacing $l \rightarrow 0$, with the dimensionful volume $V = (lI)^d$ held fixed. Accordingly, the sum $l^d \sum_{\mathbf{i}}$ over lattice sites turns into the integral $\int d^d x$. In order to maintain the correct dimensional properties, one must rescale the fields by a certain power of the lattice spacing $l$. In the quantum formulation both fields are rescaled in the same way

$$\frac{a_{\mathbf{i}}}{l^{d/2}} \rightarrow \psi(\vec{x}), \qquad \frac{a_{\mathbf{i}}^\dagger}{l^{d/2}} \rightarrow \psi^\dagger(\vec{x}). \tag{10}$$

Then, the engineering dimension of the fields $\psi(\vec{x})$ and $\psi^\dagger(\vec{x})$ is $L^{-d/2}$ with $L$ some arbitrary length unit. In this sense, the fields represent density *amplitudes*. We also note that the rescaling (10) is different from the one adopted in the classical Doi-Peliti case, briefly explained in App. A, where one rescales $\varphi$ with $l^d$ (units of a density) and the conjugated field $\bar{\varphi}$ with $l^0$. The hopping Hamiltonian (5) reads, in the continuum limit,

$$H = \int d^d x \, \mathcal{H}(\psi), \tag{11}$$

with the Hamiltonian density

$$\mathcal{H} = \psi^\dagger(\vec{x})[-J\nabla^2 + V(\vec{x})]\psi(\vec{x}). \tag{12}$$

Analogously, the dissipator (6) turns into

$$\mathcal{D}[\rho] = \int d^d x \sum_{\alpha=1}^d [L_\alpha \rho L_\alpha^\dagger - \frac{1}{2}\{L_\alpha^\dagger L_\alpha, \rho\}], \tag{13}$$

where $L_\alpha^{(\dagger)} = L_\alpha^{(\dagger)}(\vec{x})$. For one-body decay (7), the continuum limit is simply

$$L_\alpha(\vec{x}) = \frac{1}{d}\sqrt{\Gamma_d}\psi(\vec{x}). \tag{14}$$

This expression is linear in the destruction operator and therefore it produces quadratic terms in the dissipator (13). These terms can be exactly treated, as we discuss in Sec. 4. The continuum limit of the binary annihilation (8) and (9) depends on the quantum statistics. In particular, we find

$$L_\alpha^{\text{B,ann}}(\vec{x}) = \frac{1}{d}\sqrt{\Gamma}\psi^2(\vec{x}), \qquad L_\alpha^{\text{F,ann}}(\vec{x}) = \sqrt{\Gamma}\psi(\vec{x})\partial_{x_\alpha}\psi(\vec{x}). \tag{15}$$

These jump operators are quadratic in the destruction operators and therefore they produce quartic terms in the dissipator (13). These quartic terms render the Lindblad dynamics not exactly solvable. In the Keldysh field theory representation of the master equation, quartic terms amount to quartic interaction vertices in the fields of the theory. Crucially, in the fermionic case, quartic interaction vertices also contain spatial gradients of the fields, which are absent in the bosonic case. This important difference for $A + A \to \emptyset$ eventually renders the physical macroscopic behaviour of the Fermi gas different to the one of the Bose gas.

## 3 Keldysh field theory of reaction-diffusion models

In this Section, we briefly review some aspects of the Keldysh path integral formalism for the quantum master equation. These aspects are necessary for the understanding of the results presented in Secs. 4 and 5. In Subsec. 3.1, we summarise from Refs. [64–67] the closed-time contour characterising the nonequilibrium Keldysh action. Bosonic and fermionic Green's functions are building block of the Keldysh field theory and they are discussed in Subsec. 3.2. In Subsec. 3.3, we discuss the regularisation of the interaction vertices in the Keldysh action which must be considered when treating open systems.

### 3.1 Keldysh formalism for open systems

Starting from the formal solution of the Lindblad equation, the first step to derive a field theory is to perform a Trotter decomposition of the evolution operator. It is now convenient to consider the set of bosonic [fermionic] coherent states $|\{\psi\}_n\rangle = \otimes_{\mathbf{i}}|\psi_{\mathbf{i},n}\rangle$ at the time slice $t_n$, defined as the eigenstates of the annihilation operator, namely:

$$a_{\mathbf{i}}|\psi_{\mathbf{i},n}\rangle = \psi_{\mathbf{i},n}|\psi_{\mathbf{i},n}\rangle. \tag{16}$$

This entails the explicit definition

$$|\psi_{\mathbf{i},n}\rangle = e^{\xi\,\psi_{\mathbf{i},n}a_{\mathbf{i}}^\dagger}|0\rangle, \tag{17}$$

with $|0\rangle$ the vacuum state in the many-body Fock space. It is here important to stress that albeit we use the same symbol $\psi$ for the eigenvalue both in the fermions and in the bosons case, the values $\psi$ takes are different in the two cases. For bosons $\psi \in \mathbb{C}$ is a complex number, and $\bar{\psi}$ is the associated complex conjugate. For fermions, instead, $\psi$ is a Grassmann field belonging to an anticommuting algebra: $[\psi_{\mathbf{i},n}, \psi_{\mathbf{j},n}]_+ = 0$, $\forall \mathbf{i}, \mathbf{j}$, with the Grassmann field $\bar{\psi}$ independent from $\psi$. The resolution of the identity at time step $n$ takes the form:

$$\mathbb{1}_n = \frac{1}{\sqrt{\pi}^{1+\xi}}\int\prod_{\mathbf{i}}d\bar{\psi}_{\mathbf{i},n}d\psi_{\mathbf{i},n}e^{-\sum_{\mathbf{i}}\bar{\psi}_{\mathbf{i},n}\psi_{\mathbf{i},n}}|\{\psi\}_n\rangle\langle\{\psi\}_n|. \tag{18}$$

(a)

$$e^{-iH\delta t}$$

$t$                  $t_0$

$+\infty \longleftarrow$    $-\infty$

$|\psi(t)\rangle$       $\delta t$     $|\psi(t_0)\rangle$

(b)

$$e^{\mathcal{L}\delta t}$$

$t$   + contour        $t_0$

$+\infty \longleftarrow$    $-\infty$

$\rho(t)$    − contour    $\delta t$   $\rho(t_0)$

Figure 1: **Out-of-equilibrium time integration contour.** (a) Time evolution of a wave function $|\psi(t)\rangle$ along a single, forward-time integration contour from $t_0$ to $t$. (b) Time evolution of a density matrix $\rho(t)$ for the open-system out-of-equilibrium dynamics generated by the Lindbladian superoperator $\mathcal{L}$. The integration of the ensuing Keldysh actions is performed along a closed time contour, determined by the forward (+) and the backward (-) branch between times $t_0$ and $t$. The two branches are connected at the initial $t_0$ and final time $t$ of the dynamics.

The above completeness relation can be inserted on both sides of the density matrix $\rho_n$, using the superscript $\left|\{\psi\}_n^+\right\rangle$ for states acting from the left, and $\left|\{\psi\}_n^-\right\rangle$ for states acting from the right:

$$
\rho_n = \frac{1}{\pi^{1+\xi}} \int \prod_{\mathbf{i}} d\psi_{\mathbf{i},n}^+ d\bar{\psi}_{\mathbf{i},n}^+ d\psi_{\mathbf{i},n}^- d\bar{\psi}_{\mathbf{i},n}^-
$$
$$
\cdot e^{-\sum_{\mathbf{i}} \bar{\psi}_{\mathbf{i},n}^+ \psi_{\mathbf{i},n}^+} e^{-\sum_{\mathbf{i}} \bar{\psi}_{\mathbf{i},n}^- \psi_{\mathbf{i},n}^-} \left|\{\psi\}_n^+\right\rangle \left\langle \xi\{\psi\}_n^-\right| \left\langle\{\psi\}_n^+\right| \rho_n \left|\xi\{\psi\}_n^-\right\rangle.
\tag{19}
$$

Accordingly, the eigenvalues $\{\psi\}_n^+$ of $\left|\{\psi\}_n^+\right\rangle$ will identify the *forward* fields, as they evolve the density matrix forward in time, when reading from right to left. The eigenvalues $\{\psi\}_n^-$ of $\left|\{\psi\}_n^-\right\rangle$ will instead define the *backward* fields, which evolve the state backward in time as they must be read from left to right. A pictorial representation of the forward and backward fields, and of the closed Keldysh integration contour, is given in Fig. 1.

Multiplying times the identity, we connect the $\rho_n$ to $\rho_{n+1}$ at the subsequent time slices via an element-wise notation. In particular, the Lindbladian dynamics is rendered by means of the "supermatrixelement" $\left\langle \psi_{n+1}^+ \right| \mathcal{L}\left[ \left|\psi_n^+\right\rangle \left\langle \psi_n^-\right| \right] \left|\psi_{n+1}^-\right\rangle$. It is now useful to recall that expectation values of normal-ordered functions of ladder operators acting on coherent states turn into functions of the respective coherent states eigenvalues:

$$
\langle\{\psi\}_n| M(a_{\mathbf{i}}, a_{\mathbf{j}}^\dagger) |\{\psi'\}_m\rangle = e^{\sum_{\mathbf{k}} \bar{\psi}_{\mathbf{k},n} \psi'_{\mathbf{k},m}} M(\bar{\psi}_{\mathbf{i},n}, \psi'_{\mathbf{j},m}).
\tag{20}
$$

We assume henceforth the Hamiltonian $H$, in Eq. (12), and the jump operators $L_\alpha(\vec{x})$ and $L_\alpha^\dagger(\vec{x})$, in Eq. (13), to be normal-ordered, so as all the creation operators lie on the left of the destruction operators. Applying this substitution and exponentiating, it is possible to write an expression for the density matrix element at time $n+1$, where second order terms in $\delta t$ are suppressed. By taking the continuum time limit $N \to \infty$, $\delta t \to 0$, with $N\delta t = t - t_0$, the terms with $(\bar{\psi}_{\mathbf{i},n+1}^+ - \bar{\psi}_{\mathbf{i},n}^+)\psi_{\mathbf{i},n}^+$ $[\bar{\psi}_{\mathbf{i},n}^-(\psi_{\mathbf{i},n+1}^- - \psi_{\mathbf{i},n}^-)]$ can be written as $\delta t\, \bar{\psi}_{\mathbf{i},n}^+ \partial_t \psi_{\mathbf{i},n}^+$ $[\delta t\, \bar{\psi}_{\mathbf{i},n}^- \partial_t \psi_{\mathbf{i},n}^-]$. Similarly, the Trotter decomposition of the evolution operator, initially a sum of finite time slices $\sum_n \delta t$, turns into the integral $\int dt'$. Besides, fields at adjacent time slices $n$, $n+1$ can be evaluated at the same time $t' = t_0 + n\delta t$, $n = 1, 2 \ldots N$. In the continuum picture, we shall also introduce the symbol $\mathbf{D}\psi_\mu$, indicating infinite-dimensional integration over the possible field configurations, each of which must be considered at each infinitesimal Trotter time slice, namely:

$$
\mathbf{D}\psi_\mu = \lim_{I,N\to\infty} \prod_{n=0}^{N} \prod_{\mathbf{i}=0}^{I} \frac{d\psi_{\mathbf{i},n}^{\mu}}{\sqrt{\pi}^{1+\xi}},
\tag{21}
$$

with $\mu = +,-$. We eventually evolve from the initial to the final time, noticing that the $+$ and $-$ fields are connected at the boundaries. We now take the trace of the density matrix at the

final time argument $n = N$ defining the *partition function* $Z = \mathrm{tr}\,\rho(t)$:

$$Z = \int \mathbf{D}[\psi_+, \bar{\psi}_+, \psi_-, \bar{\psi}_-] e^{-\int_x \bar{\psi}_+(t_0)\psi_+(t_0)} \langle \{\psi_+(t_0)\} | \rho(t_0) | \xi\{\psi_-(t_0)\} \rangle \, e^{iS[\psi_+, \bar{\psi}_+, \psi_-, \bar{\psi}_-]}, \quad (22)$$

with $\int_x = \int d^d x$. We note that in the path integral in Eq. (22) boundary terms are present, where the fields are evaluated at the initial time $t_0$. The bulk functional $S[\psi_+, \bar{\psi}_+, \psi_-, \bar{\psi}_-]$ is named the *Keldysh action*, and it can be conveniently written as

$$S = \int_{t_0}^t dt' \int dx [\bar{\psi}_+ i \partial_{t'} \psi_+ - \bar{\psi}_- i \partial_{t'} \psi_- - i\mathcal{L}], \quad (23)$$

where we dropped the explicit dependence on space-time variables $\psi_\mu(\vec{x}, t') = \psi_\mu$, with the Lindbladian $\mathcal{L}(\psi_+, \bar{\psi}_+, \psi_-, \bar{\psi}_-)$ evaluated in terms of coherent states:

$$\mathcal{L} = \frac{\mathcal{H}_+ - \mathcal{H}_-}{i\hbar} + \left[ (L_+ \bar{L}_-)^{(1+\xi)/2} + (\bar{L}_- L_+)^{(1-\xi)/2} - 1 - \frac{1}{2}(\bar{L}_+ L_+ + \bar{L}_- L_-) \right]. \quad (24)$$

When the final $t \to +\infty$ and initial $t_0 \to -\infty$ times are sent to infinity, one focuses on the stationary state properties of the dynamics. Besides, initial-time boundary terms are neglected, as they do not affect the stationary state, and one obtains the *Keldysh partition function* $Z_K$:

$$Z_K = \int \mathbf{D}[\psi_+, \bar{\psi}_+, \psi_-, \bar{\psi}_-] e^{iS[\psi_+, \bar{\psi}_+, \psi_-, \bar{\psi}_-]}. \quad (25)$$

The Keldysh partition function obeys the normalisation $Z_K = 1$, which follows from the trace-preservation property of the Lindblad dynamics. The Keldysh partition function $Z_K$ therefore carries no memory of the initial state, which is contained in the boundary terms of Eq. (22). In the case of our quantum RD models, however, the interesting dynamics manifests in the approach towards the stationary state. We will therefore consider in Sec. 4 the whole partition function (22) in order to compute how correlation functions actually depend on the initial-boundary terms.

The Hamiltonian $\mathcal{H}_\pm = \mathcal{H}(\bar{\psi}_\pm, \psi_\pm)$ and the jump operators $L_{j,+} = L_j(\bar{\psi}_+, \psi_+)$, $L_{j,-} = L_j(\bar{\psi}_-, \psi_-)$, $\bar{L}_{j,+} = L_j^\dagger(\bar{\psi}_+, \psi_+)$ and $\bar{L}_{j,-} = L_j^\dagger(\bar{\psi}_-, \psi_-)$, after normal ordering, are evaluated on the forward (+) and backward (−) contour, respectively. In the quantum RD case of the quadratic hopping Hamiltonian (5), the Hamiltonian density Eq. (12) evaluated in terms of fields $\psi_\pm$, $\bar{\psi}_\pm$ reads as:

$$\mathcal{H}_\pm = \bar{\psi}_\pm(-J\nabla^2 + V)\psi_\pm. \quad (26)$$

It is convenient at this point to introduce the so-called Keldysh rotation [65] of the bosonic field variables, defined by:

$$\psi_\mu = \frac{\phi_c + \mu\phi_q}{\sqrt{2}}, \qquad \bar{\psi}_\mu = \frac{\bar{\phi}_c + \mu\bar{\phi}_q}{\sqrt{2}}, \quad (27)$$

with $\mu = +, -$. The new fields $\phi_c$, $\phi_q$ are called *classical* and *quantum* fields, respectively. In classical-quantum basis, named henceforth the *retarded-advanced-Keldysh* (RAK) basis, the Hamiltonian term $\mathcal{H}_+ - \mathcal{H}_-$ in the Lindbladian (24), with the quadratic $\mathcal{H}_\pm$ of Eq. (26) is given by a sum $\mathcal{H} = \mathcal{H}_c + \mathcal{H}_q$, where $\mathcal{H}_{c/q}$ is defined by

$$\mathcal{H}_{c/q} = \bar{\phi}_{q/c}(-J\nabla^2 + V)\phi_{c/q}. \quad (28)$$

For the Keldysh rotation of fermionic fields, we follow the convention of Ref. [78], namely:

$$\psi_\mu = \frac{\phi_1 + \mu\phi_2}{\sqrt{2}}, \qquad \bar{\psi}_\mu = \frac{\bar{\phi}_2 + \mu\bar{\phi}_1}{\sqrt{2}}, \quad (29)$$

with $\mu = +, -$. Note that this convention for the rotation of fermionic fields is different from that adopted for bosons (27). We also avoid for fermions the $c, q$ notation, and use indices 1, 2, as no classical behaviour is associated to fermions. In the RAK basis, the Hamiltonian term $\mathcal{H} = \mathcal{H}_+ - \mathcal{H}_-$ turns into $\mathcal{H}_1 + \mathcal{H}_2$

$$\mathcal{H}_{1/2} = \bar{\phi}_{1/2}(-J\nabla^2 + V)\phi_{1/2}. \tag{30}$$

The Keldysh rotation is extremely useful in order to get rid of redundant degrees of freedom in the Green's functions associated to the field theory (23)-(30). We elucidate this aspect in the next Subsection.

## 3.2 Green's functions

In the Keldysh field theory, one defines four two-point Green's functions in the $\pm$ basis. We thus use henceforth a convenient space-time notation for the position variable $x = (\vec{x}, t)$, so as Green's functions are conveniently written as

$$\hat{G}^{\pm}(x_1, x_2) = \begin{pmatrix} G^T(x_1, x_2) & G^<(x_1, x_2) \\ G^>(x_1, x_2) & G^{\tilde{T}}(x_1, x_2) \end{pmatrix} = -i\begin{pmatrix} \langle \psi_+(x_1)\bar{\psi}_+(x_2) \rangle & \langle \psi_+(x_1)\bar{\psi}_-(x_2) \rangle \\ \langle \psi_-(x_1)\bar{\psi}_+(x_2) \rangle & \langle \psi_-(x_1)\bar{\psi}_-(x_2) \rangle \end{pmatrix}, \tag{31}$$

which take the names of time-ordered $G^T$, lesser $G^<$, greater $G^>$, and anti-time-ordered $G^{\tilde{T}}$ correlation functions. The reason follows from the fact that on the closed integration contour in Fig. 1, the backward fields follow in time the forward ones. In particular, Green's functions can be connected to expectation values of time-ordered correlation functions of the second-quantised field operators $a(x)$, $a^{\dagger}(x)$, where time-ordering is performed along the Keldysh contour [64–67]. Their expressions read

$$iG^T(x_1, x_2) = \langle T[a(x_1)a^{\dagger}(x_2)] \rangle, \tag{32a}$$

$$iG^<(x_1, x_2) = \xi \langle a^{\dagger}(x_2)a(x_1) \rangle, \tag{32b}$$

$$iG^>(x_1, x_2) = \langle a(x_1)a^{\dagger}(x_2) \rangle, \tag{32c}$$

$$iG^{\tilde{T}}(x_1, x_2) = \langle \tilde{T}[a(x_1)a^{\dagger}(x_2)] \rangle, \tag{32d}$$

where $\Theta(t_1 - t_2)$ is the Heaviside theta function. In the previous equation, $T$, and $\tilde{T}$ denote time and anti-time ordering along the Keldysh contour, respectively. For time ordering, the operator at the latest time goes to the left, while for anti-time ordering it goes to the right. In the case of fermions, a minus sign is also added for each permutation necessary to bring the operators to the desired order. In the previous equation (32), all the operators appearing in the expectation value are meant in the Heisenberg representation. The Heisenberg representation of dynamical two-point correlation functions, where the two operators are placed at different times, in the dissipative setup is not trivial, and we refer the reader to Section 5.2 of Ref. [79] for a detailed discussion. Expressions in Eq. (32) make it evident that the equal-time evaluation of the Green's functions allows us to calculate the particle density, via the relation $\langle n(\vec{x}, t) \rangle = \langle a^{\dagger}(x)a(x) \rangle$. The $\pm$ basis, though directly constructed from the closed contour functional integral, contains a large degree of redundancy. From Eq. (32), one, indeed, sees that the same quantum mechanical operator can correspond to different Green's functions. In fact, from Eq. (32) it follows that not all the Green's functions are independent of each other:

$$G^T + G^{\tilde{T}} - G^< - G^> = 0. \tag{33}$$

It is then convenient to use the Keldysh rotation Eqs. (27) (bosons) and (29) (fermions). Then, the expressions for the Green's functions are given in the bosonic $cq$ basis by:

$$\hat{G}_B^{RAK}(x_1, x_2) = \begin{pmatrix} G^K(x_1, x_2) & G^R(x_1, x_2) \\ G^A(x_1, x_2) & 0 \end{pmatrix} = -i\begin{pmatrix} \langle \phi_c(x_1)\bar{\phi}_c(x_2) \rangle & \langle \phi_c(x_1)\bar{\phi}_q(x_2) \rangle \\ \langle \phi_q(x_1)\bar{\phi}_c(x_2) \rangle & \langle \phi_q(x_1)\bar{\phi}_q(x_2) \rangle \end{pmatrix}, \tag{34}$$

(a) $\quad \hat{G}_0(x_1, x_2)$ $\qquad$ (b) $\quad G_0^R(x_1, x_2)$ $\qquad$ (c) $\quad G_0^A(x_1, x_2)$ $\qquad$ (d) $\quad G_0^K(x_1, x_2)$

$x_1 \longleftarrow\!\!\longleftarrow x_2$ $\quad x_1 \longleftarrow\!-\!\blacktriangleleft\!-\!-\!- x_2$ $\quad x_1 -\!-\!\blacktriangleleft\!-\!-\!-\!\longleftarrow x_2$ $\quad x_1 \longleftarrow\!\!\longleftarrow x_2$

Figure 2: **Diagrammatic representation of Green's functions.** (a) Feynman diagram of the Keldysh matrix $\hat{G}_{B,F}^{RAK}$ of propagators in Eqs. (34) and (35). (b)-(c)-(d) Feynman diagrams of the retarded $G_0^R$ (b), advanced $G_0^A$ (c), and Keldysh $G_0^K$ (d) propagators, respectively. For all graphs, the leftmost lines are always associated to fields $\phi_\mu$ entering the vertex $x_1$, and the rightmost lines depict the fields $\bar{\phi}_\mu$ exiting the vertex $x_2$, with $\mu = c, q$ (bosons), $\mu = 1, 2$ (fermions). In the bosonic case, solid lines represent classical fields $\phi_c, \bar{\phi}_c$, whereas dashed lines represent quantum fields $\phi_q, \bar{\phi}_q$. In the fermionic case, solid lines represent fields $\phi_1, \bar{\phi}_2$, whereas dashed lines represent fields $\phi_2, \bar{\phi}_1$. In the figure, the subscript 0, in $G_0^{R,A,K}$ refers to the fact that the displayed Green's functions are the bare ones, i.e., they are associated to the quadratic part of the Keldysh action (see the discussion in Sec. 4).

where the indices R, A, K stand for retarded, advanced, Keldysh, respectively. For fermions the RAK basis reads as:

$$\hat{G}_F^{RAK}(x_1, x_2) = \begin{pmatrix} G^R(x_1, x_2) & G^K(x_1, x_2) \\ 0 & G^A(x_1, x_2) \end{pmatrix} = -i \begin{pmatrix} \langle \phi_1(x_1)\bar{\phi}_1(x_2) \rangle & \langle \phi_1(x_1)\bar{\phi}_2(x_2) \rangle \\ \langle \phi_2(x_1)\bar{\phi}_1(x_2) \rangle & \langle \phi_2(x_1)\bar{\phi}_2(x_2) \rangle \end{pmatrix}. \quad (35)$$

The advantage of the RAK basis is now made explicit since the redundancy of the Green's functions is removed. As a matter of fact, for bosons, the $qq$ entry in Eq. (34) is identically vanishing $\langle \phi_q(x_1)\bar{\phi}_q(x_2) \rangle \equiv 0$. For fermions, similarly, the entry (2, 1) of (35) is identically zero $\langle \phi_2(x_1)\bar{\phi}_1(x_2) \rangle = 0$. Furthermore, this definition neatly identifies the physical meaning of the three Green's functions. The *retarded* and *advanced* Green's functions $G^{R,A}$ are response functions defining the spectral properties of the quasiparticle modes of the many-body system. Conversely, the Keldysh Green's function $G^K$ carries information on the statistical occupation of quasiparticle modes and it depends on the initial distribution. Indeed, in the operatorial formalism the Green's functions $G^{R,A,K}$ read as

$$iG^R(x_1, x_2) = \Theta(t_1 - t_2)\langle [a(x_1), a^\dagger(x_2)]_\xi \rangle, \quad (36a)$$

$$iG^A(x_1, x_2) = -\Theta(t_2 - t_1)\langle [a(x_1), a^\dagger(x_2)]_\xi \rangle, \quad (36b)$$

$$iG^K(x_1, x_2) = \langle [a(x_1), a^\dagger(x_2)]_{-\xi} \rangle. \quad (36c)$$

Clearly, $G^K$ is connected to the particle density via evaluation at equal space-time points $x_1 = x_2 = x$:

$$iG^K(\vec{x}, t, \vec{x}, t) = 2\xi \langle n(\vec{x}, t) \rangle + \langle [a(x), a^\dagger(x)]_\xi \rangle. \quad (37)$$

We note that the second term $\langle [a(x), a^\dagger(x)]_\xi \rangle$ on the right hand side of (37) is divergent in the infinite volume limit (it is proportional to a Dirac delta of zero argument). This divergence can be regularised by introducing an ultraviolet – short-distance – cutoff. In Fig. 2, we report a diagrammatic representation of the Green's functions $G^{R,A,K}$, which will be used in the derivation of the results of Secs. 4 and 5.

We list here few important properties of the Green's functions, which will be used in the next Sections. First, the following Hermitian conjugation properties define the so-called "causal structure" [65] of Keldysh theory:

$$\left[ G^K(x_1, x_2) \right]^\dagger = \left[ G^K(x_2, x_1) \right]^* = -G^K(x_1, x_2), \quad (38a)$$

$$\left[ G^R(x_1, x_2) \right]^\dagger = \left[ G^R(x_2, x_1) \right]^* = G^A(x_1, x_2). \quad (38b)$$

Here the adjoint amounts to complex conjugation plus swap of space-time indices. It is possible to parameterise the anti-Hermitian $G^K$ in terms of the $G^R$ and the $G^A$:

$$G^K(x_1, x_2) = G^R \circ F - F \circ G^A. \tag{39}$$

Here, we introduced the notation $A \circ B$, where $\circ$ denotes a space-time convolution, namely

$$(A \circ B)(x_1, x_2) = \int dx_3 A(x_1, x_3) B(x_3, x_2). \tag{40}$$

The function $F$ is called the bosonic (fermionic) *distribution function* and is a Hermitian function, namely:

$$\left[ F(x_1, x_2) \right]^\dagger = \left[ F(x_2, x_1) \right]^* = F(x_1, x_2). \tag{41}$$

Physical information can be easily extracted from the Green's functions considering the set of Wigner coordinates

$$x = \frac{x_1 + x_2}{2}, \qquad x' = x_1 - x_2, \tag{42}$$

with the inverse change of coordinate $x_1 = x + x'/2$, $x_2 = x - x'/2$. The *Wigner transform* of the Green's functions is a Fourier transform in the relative space-time coordinate $x'$ [80–82], which introduces a conjugated momentum-frequency vector $k = (\vec{k}, \epsilon)$:

$$A(x, k) = \int d^d x' dt' e^{-i(\vec{k} \cdot \vec{x}' - \epsilon t')} A\left(x + \frac{x'}{2}, x - \frac{x'}{2}\right) = \int dx' e^{-ikx'} A\left(x + \frac{x'}{2}, x - \frac{x'}{2}\right), \tag{43}$$

with $kx' = \vec{k} \cdot \vec{x}' - \epsilon t$. In Appendix B, we review the fundamental properties of the Wigner transform needed for the analysis. Wigner-transforming in space $\vec{x}'$ the equal-time ($t' = 0$) Keldysh Green's function (37), we establish a direct connection with the phase-space $(\vec{x}, \vec{k})$ occupation function $n(\vec{x}, t, \vec{k})$:

$$i G^K(\vec{x}, t, \vec{k}, 0) = 1 + 2\xi n(\vec{x}, t, \vec{k}). \tag{44}$$

Here, $n(\vec{x}, t, \vec{k})$ is known as the one-body Wigner function, i.e., the semiclassical quasidistribution function [83–85]. The *spectral function* $A(x, k)$ (in the Wigner coordinates) is directly related to the retarded Green's function $G^R(x, k)$ through the relation

$$A(x, k) \equiv i[G^R(x, k) - G^A(x, k)] = -2 \operatorname{Im} G^R(x, k), \tag{45}$$

where the second identity follows from Eq. (38). The spectral function gives information about the quasiparticle spectrum of the model and it is sharply peaked around the quasiparticle dispersion relation, as long as quasiparticle excitations are well-defined. We will exploit this aspect in the derivation of the kinetic equation for $A + A \to \emptyset$ of Sec. 5.

## 3.3 Regularisation of tadpole diagrams

The perturbative expansion of the interaction vertices of the Keldysh field theory possibly leads to ill-defined diagrams. In the present manuscript, a relevant class of such diagrams is given by *tadpole* graphs. The latter are diagrams involving contractions of fields connected to the same vertex, which entail the evaluation of equal-space-time $x_1 = x_2$ propagators in Fig. 2. This leads to ill-defined quantities since $G^{R,A}(0)$ are ill-defined for equal time arguments due to the ambiguity $\Theta(0)$ (cf. Eqs. (36a) and (36b)). It is thus necessary to introduce a regularisation scheme for these diagrams, which will be relevant in Sec. 5 for the interacting theory of binary annihilation $A + A \to \emptyset$.

We will do this by introducing a temporal regularisation in the Keldysh action (23) in the form of an infinitesimal time shift $\varepsilon > 0$. The origin of this time shift can be understood by heading back to the discrete-time formulation of the Keldysh action of Subsec. 3.1, as explained in Refs. [67, 86]. Equal-time arguments arise, indeed, because of the continuum-time limit $\delta t \to 0$ after Eq. (20), but are absent in a time-discrete picture. In fact, operators in Eq. (20) are evaluated against coherent states at adjacent (but different) time slices in the construction of the path integral via the Trotter decomposition. Consequently, the terms $\bar{L}_+(t)L_+(t)$ $[\bar{L}_-(t)L_-(t)]$ come from the expectation $\langle \psi_{n+1}^+|L^\dagger L|\psi_n^+\rangle$ $[\langle \psi_n^-|L^\dagger L|\psi_{n+1}^-\rangle]$, where coherent states appear with increasing [decreasing] time arguments from the right to the left.[1] The following condition then finds a natural justification:

$$\bar{L}_+(t)L_+(t) \to \bar{L}_+(t)L_+(t-\varepsilon), \tag{46a}$$

$$\bar{L}_-(t)L_-(t) \to \bar{L}_-(t)L_-(t+\varepsilon). \tag{46b}$$

Conversely, no prearranged convention can be identified for the time direction of operators $\bar{L}_-(t)L_+(t)$ and $L_+(t)\bar{L}_-(t)$ since they couple operators $L$ and $L^\dagger$ evaluated on the different forward and backwards contours. A beneficial choice follows from probability conservation and symmetry with respect to the time shift $\varepsilon$, i.e., by requiring that the ensuing Keldysh action vanishes when dropping $\pm$ indices for any of $\varepsilon$, and that the regularised expression reduces to the original one in the limiting case $\varepsilon \to 0$. Hence, under these assumptions, we find the symmetric regularisation:

$$\bar{L}_-(t)L_+(t) \to \frac{1}{2}\bar{L}_-(t)L_+(t+\varepsilon) + \frac{1}{2}\bar{L}_-(t)L_+(t-\varepsilon). \tag{47}$$

Of course, this structure must also be carried over to the *RAK* basis, so that the interaction vertices in the *RAK* basis can be still distinguished in terms of the $(t \pm \varepsilon)$ regularisation. We do this in Sec. 5, where the interacting Keldysh field theory associated to $A + A \to \emptyset$ is considered. We will consider $\varepsilon$ to be finite in order to compute equal-time Green's functions: the latter will be vanishing (if $G^{\mu\nu}(\varepsilon) \sim \Theta(-\varepsilon) = 0$) or nonzero (if $G^{\mu\nu}(\varepsilon) \sim \Theta(\varepsilon)$). After this, we will be eventually able to safely set $\varepsilon = 0$ in the expressions resulting from the tadpole diagrams.

## 4 Initial-time boundary conditions

In this Section, we study the Keldysh partition function (22) containing both the boundary initial terms and the bulk Keldysh action $S$. We set in Eq. (22) the final time $t \to \infty$, and the initial time $t_0 = 0$. This allows us to study correlation functions in the time interval $[0, +\infty]$ without losing information on the initial state, and therefore to eventually go beyond the stationary state physics. We will consider both pure states, with a prescribed mean initial density, and mixed thermal-Gibbs states. In Subsec. 4.1, we study the simple case of one-body decay $A \to \emptyset$ for bosons (14) (the calculations easily generalises to the fermionic case) in an initial pure coherent state. This is an exactly solvable case which allows to benchmark the effect of the boundary initial term in the Keldysh action. In particular, we find that boundary terms at $t_0 = 0$ are necessary in order to obtain the correct dynamical approach of the density towards the steady state devoid of particles. The same analysis is performed in Subsec. 4.2 for the case of one-body $A \to \emptyset$ decay for bosons from thermal initial states. In Subsec. 4.3, we show how disconnected diagrams in the perturbative expansion of the boundary terms generate the normalisation of the Keldysh partition function.

---

[1]This reasoning assumes that $L^\dagger L$ is normal ordered if $L$ and $L^\dagger$ are. This is true for Eq. (15), but it is not true in general. In the latter case, one needs to insert one additional resolution of the identity between $L$ and $L^\dagger$ in writing the coherent-state path integral, cf. the discussion in Appendix A2 of Ref. [67].

## 4.1 Coherent initial state

We consider an initial pure state

$$|\Psi_0\rangle = \frac{1}{\sqrt{\mathcal{N}}} \prod_{\mathbf{i}} |\Psi\rangle_{\mathbf{i}}, \qquad |\Psi\rangle_{\mathbf{i}} = \sum_{n_{\mathbf{i}}=0}^{\infty} \frac{\sqrt{\tilde{n}_0}^{n_{\mathbf{i}}}}{\sqrt{n_{\mathbf{i}}!}} |n_{\mathbf{i}}\rangle, \tag{48}$$

with the normalisation factor $\mathcal{N} = \prod_{\mathbf{i}} e^{\tilde{n}_0}$. The state $|\Psi_0\rangle$ is normalised $\langle \Psi_0 | \Psi_0 \rangle = 1$ and it is a tensor product of coherent states $|\Psi\rangle_{\mathbf{i}}$ at each lattice site $\mathbf{i}$. The initial average particle number is $\tilde{n}_0$ on each lattice site (the state is translational invariant). The state Eq. (48) can be interpreted as the quantum analogue of the initial states considered in the field theory of classical RD systems [8–10] (see also Appendix A). In the classical case, indeed, each lattice site is occupied by a number of particles distributed according to a Poissonian probability. In the quantum case, the Poissonian distribution is obtained squaring the amplitudes in the state $|\Psi_0\rangle$. We also note that the state (48) has a clear interpretation in momentum space (see Eq. (66) below for the definition of Fourier transformed operators). In particular, using the definition (17), the initial state (48) is recognised as a coherent state of the mode $k = 0$

$$|\Psi_0\rangle = \frac{1}{\mathcal{N}} \exp(\sqrt{\tilde{n}_0 V} \, \hat{b}_{k=0}^{\dagger}) |0\rangle = \frac{1}{\mathcal{N}} \exp(\sqrt{N} \, \hat{b}_{k=0}^{\dagger}) |0\rangle. \tag{49}$$

This representation of the initial state makes transparent that initially only the mode $k = 0$ is occupied. This state is therefore akin to a Bose-Einstein condensate, which for a macroscopic occupation $N \gg 1$ of the mode $k = 0$ can be, indeed, represented within the Bogoliubov approximation [87] with a coherent state of the mode $k = 0$.

The matrix element of the state $\rho(t_0) = |\Psi_0\rangle \langle \Psi_0|$ in the coherent state basis at time $t_0 = 0$ is

$$\langle \{\psi_+(t_0)\} | \rho(t_0) | \xi \{\psi_-(t_0)\} \rangle = \frac{1}{\mathcal{N}} \prod_{\mathbf{i}} \exp\left\{ \sqrt{\tilde{n}_0} [\bar{\psi}_{+,\mathbf{i}}(t_0) + \xi \psi_{-,\mathbf{i}}(t_0)] \right\}. \tag{50}$$

Taking the space continuum limit and rescaling the fields $\psi_{\pm}$ and $\bar{\psi}_{\pm}$ according to Eq. (10) and the particle density as $n_0 = \tilde{n}_0/l^d$ we obtain the boundary term ($t_0 = 0$)

$$\langle \{\psi_+(0)\} | \rho(0) | \xi \{\psi_-(0)\} \rangle = \frac{1}{\mathcal{N}} \exp\left\{ \sqrt{n_0} \int_0^{\infty} dt \, \delta(t) \int_x \left( \bar{\psi}_+ + \psi_- \right) \right\}, \tag{51}$$

which must be inserted in the action according to Eq. (22). The normalisation factor $\mathcal{N}$, which in the continuum limit reads $\mathcal{N} = \exp(\int_x n_0)$, is crucial in order to maintain the trace-preservation property $Z = Z_K = 1$. When $\mathcal{N}$ is not included in the Keldysh action, the calculation of physical Green's functions will have to take into account a different normalisation, i.e., $Z = \mathcal{N}$.

As a first benchmark, we consider one-body decay $A \to \emptyset$ as in Eq. (14) for the bulk Keldysh action. This case is exactly solvable yielding an exponential decay in time of the particle density. We discuss the bosonic case for the sake of illustration purposes, as the fermionic case can be worked out similarly.

The Keldysh bulk action (23) reads

$$S = \int_0^{\infty} dt \int_x d^d x \left[ \bar{\psi}_+ (i\partial_t + J\nabla^2/\hbar + \frac{i}{2}\Gamma_d)\psi_+ - \bar{\psi}_- (i\partial_t + J\nabla^2/\hbar - \frac{i}{2}\Gamma_d)\psi_- - i\Gamma_d \bar{\psi}_- \psi_+ \right]. \tag{52}$$

In addition to the bulk action (52), in the path integral (22), we have the boundary term

$$\int_x d^d x \, \bar{\psi}_+(0)\psi_+(0) = \int_0^{\infty} dt \, \delta(t) \int d^d x \, \bar{\psi}_+ \psi_+, \tag{53}$$

and the other boundary term containing the information on the initial state is given in Eq. (51). Here, we set the external potential $V(\vec{x}) = 0$. We notice that the boundary term (53) is quadratic in the fields and therefore it determines the Green's function together with the quadratic bulk action (52). The matrix of Green's functions associated to Eqs. (52) and (53) is obtained by Gaussian integration (see also the next Subsec. 4.2 for the details) and it reads:[2]

$$i\hat{G}_0^\pm(x) = \begin{pmatrix} N(\vec{x},t)e^{-\Gamma_d t/2}\Theta(t) & 0 \\ N(\vec{x},t)e^{-\Gamma_d |t|/2} & N(\vec{x},t)e^{\Gamma_d t/2}\Theta(-t) \end{pmatrix}, \tag{54}$$

with the imaginary Gaussian

$$N(\vec{x},t) = \left[\frac{i}{4\pi Jt}\right]^{d/2}\exp\left[-i\frac{x^2}{4Jt}\right], \tag{55}$$

and $x = x_1 - x_2$. Note that the Green's functions are, indeed, both space and time translation invariant $G_0^\pm(x) \equiv G_0^\pm(x_1 - x_2) = G_0^\pm(\vec{x}_1 - \vec{x}_2, t_1 - t_2)$. When $t \to 0$, the imaginary Gaussian tends to a Dirac delta $N(\vec{x},t) \to \delta(\vec{x})$. One can also check that Eq. (33) holds due to probability conservation. These Green's functions do not carry information on the system initial population, which is contained in Eq. (51).

Performing the Keldysh rotation (27), the bulk Keldysh action reads:

$$S = \int_0^\infty dt\, d^d x \left[\bar{\phi}_c(i\partial_t + J\nabla^2/\hbar - \frac{i}{2}\Gamma_d)\phi_q + \bar{\phi}_q(i\partial_t + J\nabla^2/\hbar + \frac{i}{2}\Gamma_d)\phi_c + i\Gamma_d|\phi_q|^2\right]. \tag{56}$$

The Keldysh matrix of the rotated Green's functions $G^{RAK}$ is given by the following entries:

$$i\hat{G}_0^{RAK}(x) = \begin{pmatrix} N(\vec{x},t)e^{-\Gamma_d|t|/2} & N(\vec{x},t)e^{-\Gamma_d t/2}\Theta(t) \\ -N(\vec{x},t)e^{\Gamma_d t/2}\Theta(-t) & 0 \end{pmatrix}. \tag{57}$$

Correlations between fields $\phi_q$ and $\bar{\phi}_q$ vanish as a result of probability conservation. We note that $iG_0^K(\vec{x}_1, t_1, \vec{x}_2, t_1) = \delta(\vec{x}_1 - \vec{x}_2)$, and $iG_0^K(0) = iG_0^K(x,x) \to \delta(0)$, thus yielding the bosonic statistics. This fact is consistent with Eq. (37) and it shows that the stationary density of the system is zero. This is consistent with the observation that the action (52) and (53) solely describes the stationary state of the system and therefore the associated Green's function (57) does not carry memory on the initial state. Also we note that the Keldysh component $G^K$ does not vanish, even for $\Gamma_d = 0$. This comes from the fact that we are explicitly keeping track of the boundary initial term (53), which leads to the appearance of terms $|\phi_q|^2$ in the action. In the absence of the boundary initial term (53), the Keldysh component $G^K$ vanishes for $\Gamma_d = 0$ and in order to reintroduce it, one needs to insert an infinitesimal regularisation factor in front of the term $|\phi_q|^2$ in the Keldysh action [65].

In order to account for the dynamical approach to the stationary state, we then need to include the boundary term (51). This boundary term is linear in the fields and therefore it can be included in the source term $J_\pm = (j_\pm, \bar{j}_\pm)^T$ of the generating functional $Z[J_+, J_-]$

$$Z[J_+, J_-] = \frac{1}{\mathcal{N}}\int \mathbf{D}[\psi_\pm, \bar{\psi}_\pm] e^{-\int_x \bar{\psi}_+(0)\psi_+(0)} \langle\{\psi_+(0)\}|\rho(0)|\{\psi_-(0)\}\rangle$$
$$\cdot e^{iS[\psi_+, \bar{\psi}_+, \psi_-, \bar{\psi}_-]}\exp\left\{i\int_0^t dt \int_x [J_+^\dagger\Psi_+ - J_-^\dagger\Psi_-]\right\}, \tag{58}$$

---

[2]We use the symbol $\hat{G}_0^\pm$ [$\hat{G}_0^{RAK}$] to indicate the matrix of *bare*, i.e., non-interacting, steady-state Green's function in the $\pm$ [*RAK*] basis. When interactions are introduced, we use the symbol $\hat{G}^\pm$ [$\hat{G}^{RAK}$] to indicate *dressed* steady-state Green's functions. The notation $\hat{G}_{0,S}^\pm$ [$\hat{G}_{0,S}^{RAK}$] is used for bare *physical* Green's functions including the effect of all the initial-time boundary conditions.

with the spinors $\Psi_{\pm} = (\psi_{\pm}, \bar{\psi}_{\pm})^T$. The factor $\mathcal{N}$ appearing in the denominator on the first line of the previous equation is the one normalising to unity the Poissonian initial state $\rho(t_0)$ of Eq. (48). It must be introduced by hand as it had been discarded from the definition of the boundary term (51), implying that the full partition function is now normalised to $\mathcal{N}$ (see Subsec. 4.3). In particular, from Eq. (51), one can see that the source fields are redefined as

$$j_+(x) \to j_+(x) - i\sqrt{n_0}\delta(t'), \qquad \bar{j}_- \to \bar{j}_- + i\sqrt{n_0}\delta(t'), \tag{59}$$

in order to account for the initial boundary terms (with $\bar{j}_+$ and $j_-$ unchanged). In the present case, $Z[J_+, J_-]$ can be exactly evaluated by Gaussian integration obtaining

$$Z[J_+, J_-] = \frac{1}{\mathcal{N}} \exp\left\{-i \int dx_1 \int dx_2 (\bar{j}_+(x_1), -\bar{j}_-(x_1)) G_0^{\pm}(x_1, x_2)(j_+(x_2), -j_-(x_2))^T\right\}. \tag{60}$$

Via functional differentiation of $Z[J_+, J_-]$ one calculates $iG_{0,S}^<$, with the source fields redefined as in (59). When setting all external sources $J_{\pm} = 0$ to zero, time integration is deleted by the $\delta(t)$ constraining the initial configuration, while space integration of the normalised imaginary Gaussian $N(\vec{x} - \vec{y}, t)$ of Eq. (55) gives a unit factor. This eventually allows to write the lesser Green's function as

$$iG_{0,S}^<(x_1, x_2) = \left.\frac{\delta^2 Z}{\delta\bar{j}_+(x_1)\delta j_-(x_2)}\right|_{J_{\pm}=0} = n_0 e^{-\frac{\Gamma_d}{2}(t_1+t_2)} + iG_0^<(x_1 - x_2), \tag{61a}$$

$$iG_{0,S}^<(x, x') = n_0 e^{-\Gamma_d t} + iG_0^<(x'), \tag{61b}$$

with $G_0^<(x_1 - x_2) = 0$, and where on the second line we used Wigner coordinates. Setting equal space-time coordinates $x_1 = x_2$, or, equivalently, $x' = 0$, we find

$$iG_{0,S}^<(x_1, x_1) = \left.\frac{\delta^2 Z}{\delta\bar{j}_+(x_1)\delta j_-(x_1)}\right|_{J_{\pm}=0} = n_0 e^{-\Gamma_d t_1}\Theta(t_1) + iG_0^<(0), \tag{62a}$$

$$iG_{0,S}^<(x, 0) = n_0 e^{-\Gamma_d t} + iG_0^<(0). \tag{62b}$$

The same derivation can be analogously carried on in the RAK basis. In particular, for the Keldysh Green's function $iG_{0,S}^K$ one has:

$$iG_{0,S}^K(x_1, x_2) = 2n_0 e^{-\frac{\Gamma_d}{2}(t_1+t_2)} + iG_0^K(x_1 - x_2), \tag{63a}$$

$$iG_{0,S}^K(x, x') = 2n_0 e^{-\Gamma_d t} + iG_0^K(x'), \tag{63b}$$

with $iG_0^K(x_1 - x_2) = iG_0^R(x_1 - x_2) - iG_0^A(x_1 - x_2)$ as given in Eq. (57). Both the Green's functions in Eq. (61b) and (63) are not time translational invariance since they depend not only on the relative time $t'$, but also on the centre of mass time $t$. The latter dependence is present in the first first term on the right hand side, which couples to the initial density $n_0$. Only for $\Gamma_d = 0$, time translational invariance is recovered. This comes from the fact that for $\Gamma_d = 0$, the initial state (49) is stationary with respect to the Hamiltonian evolution since only the mode $k = 0$ is populated. At equal space-time points:

$$iG_{0,S}^K(x_1, x_1) = 2n_0 e^{-\Gamma_d t_1}\Theta(t_1) + iG_0^K(0), \tag{64a}$$

$$iG_{0,S}^K(x, 0) = 2n_0 e^{-\Gamma_d t}\Theta(t) + iG_0^K(0). \tag{64b}$$

Conversely, the retarded and advanced Green's functions are not modified by the presence of initial conditions, namely $iG_{0,S}^R = iG_0^R$ and $iG_{0,S}^A = iG_0^R$, thus confirming how they only carry information concerning the steady-state properties of the systems, as we have discussed

in Subsec. 3.2. From Eqs. (62) and (64) we arrive at the expected exponentially decaying, space-independent laws for the particle density:

$$\langle n(\vec{x}, t) \rangle = n_0 e^{-\Gamma_d t}. \tag{65}$$

This calculation exemplary shows the importance of taking into account the boundary terms (51) in order to describe the full dynamics of the density. The bulk Keldysh action (52), in fact, solely describes the stationary state with zero density of particles.

## 4.2 Thermal initial state

We consider here a second relevant choice of initial conditions, which represent thermodynamic equilibrium. To do this it is convenient to introduce the Fourier transform $\hat{\psi}_k$, with momentum $k$, of the continuum-space operators $\psi(\vec{x})$ (10):

$$\hat{\psi}_k = \frac{1}{\sqrt{V}} \int_V d^d x \, e^{i\vec{k}\cdot\vec{x}} \psi(\vec{x}), \quad \text{with inverse} \quad \psi(\vec{x}) = \frac{1}{\sqrt{V}} \sum_{\vec{k}} e^{-i\vec{k}\cdot\vec{x}} \hat{\psi}_k. \tag{66}$$

In the previous equation, $\sum_{\vec{k}} = \sum_{\vec{n}}$, where $\vec{k} = 2\pi\vec{n}/I$ and $\vec{n}$ is a $d$-dimensional vector of integers which labels the allowed momenta. In each space dimension, we assume periodic boundary conditions with $I$ the associated length of the system as defined before Eq. (2). As in the previous subsection, we focus on translationally invariant systems. The initial grand canonical density matrix reads as

$$\rho(0) = \frac{1}{\mathcal{N}} e^{-\beta(H-\mu N)} = \frac{1}{\mathcal{N}} \prod_k e^{-\beta n_k (Jk^2 - \mu)}, \tag{67}$$

with $\mu$ the chemical potential associated to the total particle number $N$. The advantage of the momentum representation is that the initial density matrix is diagonal in momentum space. For a negative chemical potential, the bosonic normalisation factor is given by $\mathcal{N}_B = \text{tr}\{e^{-\beta(H-\mu N)}\} = \prod_k [1 - \exp(-\beta(Jk^2 - \mu))]^{-1}$, so that the density matrix is normalised to one. The fermionic normalisation is instead $\mathcal{N}_F = \prod_k [1 + \exp(-\beta(Jk^2 - \mu))]$. Let us now express the initial condition in terms of coherent states. It is clear that creation operators will act on state $\langle \psi_+(0)|$ on the left, while destruction operators will act on state $|\psi_-(0)\rangle$ on the right leading to the expression

$$\langle \{\psi_+(0)\} | \rho(0) | \xi \{\psi_-(0)\} \rangle = \frac{1}{\mathcal{N}} \prod_k \exp\left[ \xi \hat{\bar{\psi}}_{+,k}(0) \hat{\psi}_{-,k}(0) e^{-\beta(Jk^2 - \mu)} \right]. \tag{68}$$

We now take the infinite volume limit $V \to \infty$. In this limit, the allowed momenta $\vec{k}$ span continuously the real numbers and the matrix element of the initial state over coherent states (68) becomes

$$\langle \{\psi_+(0)\} | \rho(0) | \xi \{\psi_-(0)\} \rangle = \frac{1}{\mathcal{N}} \exp\left\{ -\int_0^\infty dt \, \delta(t) \int \frac{d^d k}{(2\pi)^d} \exp\left[-\xi\beta(Jk^2 - \mu)\right] \hat{\bar{\psi}}_+ \hat{\psi}_- \right\}. \tag{69}$$

In the previous expression we used the identity $\langle \psi | u_k^{n_k} | \psi' \rangle = e^{\bar{\psi}\psi' u_k}$, see, e.g., Ref. [65], with $u_k = e^{-\beta(Jk^2 - \mu)}$ the Boltzmann weight. The expression in Eq. (69) can be readily generalised to initial states of the GGE form $\sim \exp(-\sum_i \beta_i Q_i)$ where $Q_i$ are conserved charges of the Hamiltonian and $\beta_i$ the associated Lagrange multiplies [62, 63]. In the present case, we consider the case of a grand canonical state $Q_1 = H$ and $Q_2 = N$, for concreteness of the presentation. It is also important to note that the states (67) are, in general, mixed,

in contrast to the initial condition (48) which is a pure state. Moreover, the initial boundary term (67) is quadratic in the fields, differently from (51) which is linear. Boundary terms at time $t = 0$ which couple to quadratic expressions in the fields have been considered also in Ref. [88]. Therein generic initial states $\rho_0$ are considered and the expectation value $\langle \{\psi_+(0)\} | \rho(0) | \xi\{\psi_-(0)\} \rangle = \exp(i\delta S(u))$ is exponentiated into the Keldysh action (25) by definining the generating function $\delta S(u)$. Physical Green's functions are obtained by taking derivatives with respect to the counting parameters $u$. For the (generalised) Gibbs (67) states we do not need to introduce the generating function $\delta S(u)$ since the matrix element (69) is already exponential in the fields. This allows to compute the Green's functions directly by Gaussian integration, as we now detail. We specialise again to the case of bosons, while the generalisation to fermions is immediate.

Let us start by considering the Keldysh action in the *RAK* basis together with the boundary conditions

$$S_{\text{tot}} = S + \frac{1}{2} \int_0^\infty dt\, \delta(t) \int \frac{d^d k}{(2\pi)^d} \Big[ \hat{\bar{\phi}}_c \hat{\phi}_c (1-u_k) + \hat{\bar{\phi}}_c \hat{\phi}_q (1+u_k) + \hat{\bar{\phi}}_q \hat{\phi}_c (1-u_k) + \hat{\bar{\phi}}_q \hat{\phi}_q (1+u_k) \Big],$$
(70)

with $u_k = \exp(-\beta(Jk^2 - \mu))$ the Boltzmann weight and $S$ is the "bulk" action defined in Eq. (52), written in the momentum basis:

$$
S_{\text{tot}} = \int_0^\infty dt \int \frac{d^d k}{(2\pi)^d} \Bigg[ \hat{\bar{\phi}}_c \frac{1-u_k}{2} \delta(t) \hat{\phi}_c + \hat{\bar{\phi}}_c \left( i\partial_t - Jk^2/\hbar - \frac{i}{2}\Gamma_d + \frac{1+u_k}{2}\delta(t) \right) \hat{\phi}_q
$$
$$
+ \hat{\bar{\phi}}_q \left( i\partial_t - Jk^2/\hbar + \frac{i}{2}\Gamma_d + \frac{1-u_k}{2}\delta(t) \right) \hat{\phi}_c + \hat{\bar{\phi}}_q \left( i\Gamma_d + \frac{1+u_k}{2}\delta(t) \right) \hat{\phi}_q \Bigg]. \quad (71)
$$

One can solve the functional integral by inverting the matrix of momentum-space inverse propagators, given by:

$$(\hat{G}_{0,S}^{-1})^{RAK}(\vec{k}, t_1, t_2) = \begin{pmatrix} 0 & (G_0^{-1})^A \\ (G_0^{-1})^R & (G_0^{-1})^K \end{pmatrix} + \frac{i}{2}\delta(t_1)\delta(t_2)\delta(\vec{k}_1 - \vec{k}_2) \begin{pmatrix} 1-u & 1+u \\ 1-u & 1+u \end{pmatrix}, \quad (72)$$

with

$$(G_0^{-1})^{R/A}(\vec{k}, t_1, t_2) = \delta(\vec{k}_1 - \vec{k}_2)\delta(t_1 - t_2)[i\partial_t - Jk^2 \pm i\Gamma_d/2],$$

and

$$(G_0^{-1})^K(\vec{k}, t_1, t_2) = i\Gamma_d \delta(\vec{k}_1 - \vec{k}_2)\delta(t_1 - t_2).$$

Let us drop the explicit dependence on $\vec{k}$ for the sake of simplicity. Using the definition $(\hat{G}_{0,S}^{-1})^{RAK} \circ \hat{G}_{0,S}^{RAK} = \mathbb{1}$, we can find the following two equations for $G_{0,S}^R$:

$$\int dt_2 \frac{i}{2} \delta(t_1)\delta(t_2)(1-u_k) G_{0,S}^R(t_2, t_3) = \frac{i}{2}\delta(t_1) G_{0,S}^R(0, t_3) = 0, \quad (73a)$$

$$\int dt_2 \Big[ (i\partial_{t_1} - Jk^2 + \frac{i\Gamma_d}{2})\delta(t_1 - t_2) + \frac{i}{2}\delta(t_1)\delta(t_2)(1-u_k) \Big] G_{0,S}^R(t_2, t_3) = \delta(t_1 - t_3). \quad (73b)$$

From the first equation one has that $G_{0,S}^R(0, t_2) = 0$, consequently the second equation gives $G_{0,S}^R(t_1, t_2)$ exactly as in Eq. (57). Because of the general conjugation property $G^A(t_1, t_2) = [G^R(t_2, t_1)]^*$, $G_{0,S}^A(t_1, t_2)$ is calculated immediately and it has the same form as in Eq. (57). We therefore see that $G_{0,S}^{R,A}$ carry information only about the quasi-particle dispersion relation $\varepsilon_k = Jk^2$. On the contrary $G_{0,S}^{R,A}$ do not depend on the initial distribution $1 \pm u$, and

therefore they coincide with their stationary limit. In particular, $G_{0,S}^{R,A}(t_1, t_2) = G_{0,S}^{R,A}(t_1 - t_2)$ are time translational invariant, as expected for Green's functions describing the stationary state. Although we have shown this in the simple noninteracting case of one-body decay, this is a general property of $G^{R/A}$. In fact, in the general interacting case one can simply replace above $\pm i\Gamma_d/2 \to \Sigma^{R/A}$, with $\Sigma^{R/A}$ the retarded and advanced components of the self energy (see the definitions in the next Subsec. 5.2). Equation (73a), however, does not depend on $\Sigma^R$ implying that in general $G^R(0, t_2) = 0$. This implies the causality structure $G^{R/A}(t_1, t_2) \sim \Theta(\pm(t_1 - t_2))$. Interactions therefore dress the Green's functions $G^{R/A}$ compared to their bare values $G_{0,S}^{R/A}$ still preserving their analytic properties. The effect of the initial condition is, instead, evident on the Keldysh Green's function $G_{0,S}^K$. In particular, substituting the solution for $G_{0,S}^{R,A}$ in the equation for the advanced Green's function

$$\int dt_2 \left[ \left( i\partial_{t_1} - Jk^2/\hbar - \frac{i\Gamma_d}{2} \right) \delta(t_1 - t_2) + \frac{i}{2}\delta(t_1)\delta(t_2)(1 + u_k) \right] G_{0,S}^A(t_2, t_3)$$
$$+ \int dt_2 \frac{i}{2}\delta(t_1)\delta(t_2)(1 - u_k) G_{0,S}^K(t_2, t_3) = \delta(t_1 - t_3), \tag{74}$$

the operator $(i\partial_{t_1} - Jk^2 - \frac{i\Gamma_d}{2})G_{0,S}^A(t_1, t_3) = \delta(t_1 - t_3)$ from Eq. (73b) and therefore one obtains the desired boundary condition relating the Keldysh and the advanced Green's function at the initial time:

$$G_{0,S}^K(0, t_2) = -\frac{1 + u_k}{1 - u_k} G_{0,S}^A(0, t_2) = -(2\hat{n}_0(k) + 1) G_{0,S}^A(0, t_2). \tag{75}$$

The initial particle density $\hat{n}_0(k)$ is simply the Bose-Einstein distribution at inverse-temperature $\beta$, namely $\hat{n}_0(\vec{k}) = n_{BE}(\vec{k}) = \{\exp[\beta(Jk^2 - \mu)] - 1\}^{-1}$. Also for the Keldysh Green's function, Eq. (75) holds true in general for interacting systems. This further confirms that $G^K$, in general, keeps track of the statistical occupancy of the eigenmodes. In order to derive the expression for the full Keldysh Green's function as a function of $t_1$, we consider the last equation:

$$\int dt_2 \left[ \left( i\partial_{t_1} - Jk^2/\hbar + \frac{i\Gamma_d}{2} \right) \delta(t_1 - t_2) + \frac{i}{2}\delta(t_1)\delta(t_2)(1 - u) \right] G_{0,S}^K(t_2, t_3)$$
$$+ \int dt_2 \left[ i\Gamma_d\,\delta(t_1 - t_2) + \frac{i}{2}\delta(t_1)\delta(t_2)(1 + u) \right] G_{0,S}^A(t_2, t_3) = 0. \tag{76}$$

Substitution of the initial-time Green's functions in the latter equation yields:

$$\left[ i\partial_{t_1} - Jk^2/\hbar + \frac{i\Gamma_d}{2} \right] G_{0,S}^K(t_1, t_2) = i\Gamma_d\, G_{t_1, t_2}^A. \tag{77}$$

This equation can be solved by means of the Laplace transform in the first time variable $t_1$, which is the suitable tool to keep the information on the initial condition (75). Expressing the solution in the Wigner coordinates (42), we obtain the momentum-space Keldysh Green's function

$$iG_{0,S}^K(\vec{x}, t, \vec{k}, t') = 2\hat{n}_0(\vec{k})e^{-iJk^2 t'/\hbar} e^{-\Gamma_d t} + e^{-iJk^2 t'/\hbar} e^{-\Gamma_d|t'|/2}. \tag{78}$$

The Keldysh Green's function therefore depends on the initial distribution $\hat{n}_0(\vec{k})$. Furthermore, $G_{0,S}^K(t, t')$ is not stationary, i.e., not time translational invariant, since it depends not only on the difference $t' = t_1 - t_2$ between the two times, but also on the combination $t = (t_1 + t_2)/2$. In the stationary limit $t_{1,2} \to \infty$ and $t \to \infty$ (while $t'$ can remain finite), Eq. (78) reduces to the stationary Green's function in Eq. (57). In addition, as noted also in Subsec. 4.1 for the initial state $|\Psi\rangle_0$, we see that also at $\Gamma_d = 0$, the Keldysh Green's function is stationary. This is

expected since for zero decay the initial grand canonical state (67) is stationary with respect to the Hamiltonian evolution. Equation (78) thereby neatly shows how the inclusion of the boundary terms related to the initial state $\rho(0)$ impact on the structure of the Green's functions so that the latter encompass the whole dynamics beyond the stationary state for $\Gamma_d \neq 0$. In particular, we get the time evolution of the density by considering the equal-space $\vec{x}' = 0$ and equal-time $t' = 0$ evaluation of $G_{0,S}^K$ (cf. Eqs. (37) and (44)) which yields

$$iG_{0,S}^K(\vec{x}, t, 0, 0) = 2 \int \frac{d^d k}{(2\pi)^d} \hat{n}_0(\vec{k}) e^{-\Gamma_d t} + \delta(\vec{x}')\big|_{\vec{x}'=0} = 2 n_0 e^{-\Gamma_d t} + \delta(\vec{x}')\big|_{\vec{x}'=0}, \quad (79)$$

with $n_0$ the homogeneous initial density. From this we again conclude that the density decays exponentially in time as in Eq. (65). We note that the exponential decay does not depend on the space dimensionality $d$ since it does not couple to the space structure of the problem. Furthermore, we expect the exponential decay to apply more generically, for any initial condition, since the decay $A \to \emptyset$ does not couple different particles. In concluding, we note that the derivation explained in this Subsection can be adapted to the case when the initial boundary term (69) is disregarded. In particular, setting $u_k = 0$, one has from (75) that $G_{0,S}^K(0, t_2) = -G_{0,S}^A(0, t_2)$, which together with (77) and the result for $G_{0,S}^{R,A}(t_1, t_2)$ allows to conclude that $G_{0,S}^{RAK}(t_1, t_2)$ coincide with the expressions in Eq. (57), as anticipated in (54) and (55).

## 4.3 Perturbative expansion of the initial conditions

In this Subsection, we present an alternative approach to obtain the exponential decay in Eq. (65). This approach considers the boundary terms at $t = 0$ due to initial conditions as perturbations compared to the bulk Keldysh action (52). One then perturbatively expands the boundary terms and computes the ensuing correlations functions via Wick theorem with respect to the quadratic weight (52). This approach is similar to the one followed in Refs. [9, 10, 22] for classical RD systems (briefly recalled in Appendix A). This discussion therefore provides a first simple example of the use of perturbation theory in the Keldysh framework. Furthermore, it also allows to show how the perturbative expansion of the initial conditions correctly reproduces the normalisation $Z = \mathcal{N}$ of the Keldysh path integral (22). We will explain the method in the case of bosons, but the generalisation to fermions follows along the same lines. In the case of the Poissonian initial conditions Eq. (48), the coupling between fields and initial average density $\sqrt{n_0}$ is linear, as one can see from Eq. (51). Hence, one directly finds that in the $\pm$ basis:

$$\exp\left\{-\sqrt{n_0} \int d^d y \int_0^\infty dt_y \, \delta(t_y)[\psi_-(y) + \bar{\psi}_+(y)]\right\}$$
$$= \sum_{j=0}^\infty \frac{(-\sqrt{n_0})^j}{j!} \left\{\int d^d y [\psi_-(\vec{y}, 0) + \bar{\psi}_+(\vec{y}, 0)]\right\}^j. \quad (80)$$

Inserting the latter equation as the initial-time conditions in Eq. (22) the full partition function reads

$$Z = \sum_{j=0}^\infty \frac{(-1)^j n_0^{j/2}}{j!} \int \mathbf{D}[\psi_+, \bar{\psi}_+, \psi_-, \bar{\psi}_-] e^{iS_0} e^{-\int_x \bar{\psi}_+(t_0)\psi_+(t_0)}$$
$$\cdot \int \prod_{k=1}^j d^d y_k \prod_{k=1}^j [\psi_-(\vec{y}_k, 0) + \bar{\psi}_+(\vec{y}_k, 0)] = \sum_{j=0}^\infty Z_{(j)}, \quad (81)$$

with $S_0$ the quadratic action (52) including quantum hopping and one-body decay. All odd terms in such expansion vanish by symmetry. Thus, the partition function can be expressed as a sum of powers of $n_0$ (or even powers of $\sqrt{n_0}$). Moreover, one can see that terms of type $\psi\psi$, $\bar\psi\bar\psi$ also vanish, so that only mixed terms $\langle\psi_-(\vec{y}_1,0)\bar\psi_+(\vec{y}_m,0)\rangle_0\dots\langle\psi_-(\vec{y}_n,0)\bar\psi_+(\vec{y}_{2j},0)\rangle_0$ with $m\neq 1$, $n\neq 2j$, are nonzero. Its pictorial representation only includes disconnected diagrams where free propagators connect $\psi_-$ ($\psi_+$) fields and $\bar\psi_+$ ($\bar\psi_-$). As an example, the first non-trivial term reads

$$Z_{(2)} = \frac{n_0}{2}2\int d^d y_1 d^d y_2 \langle\psi_-(\vec{y}_1,0)\bar\psi_+(\vec{y}_2,0)\rangle_0. \tag{82}$$

Here, $\langle\dots\rangle_0$ denotes the average of the Gaussian weight given by the action (52) and (53). The physical non-interacting Green's function $G_{0,S}^<(x_1,x_2)$ computed in the presence of initial conditions, which in the operatorial formalism corresponds to the particle density, as discussed in Subsec. 3.2 for Eq. (32), is instead given by:

$$iG_{0,S}^<(x_1,x_2) = \frac{1}{Z}\sum_{j=0}^\infty \frac{(-1)^j n_0^{j/2}}{j!}\int\prod_{k=1}^j d^d y_k \cdot \langle\psi_+(x_1)\bar\psi_-(x_2)\prod_{k=1}^j[\psi_-(\vec{y}_k,0)+\bar\psi_+(\vec{y}_k,0)]\rangle_0, \tag{83}$$

where the ratio $1/Z$ indicates that the expectation value must be normalised with respect to the full partition function. Again, terms containing an odd number of fields, proportional to an odd power of $\sqrt{n_0}$, vanish. Then, the second-order correction to $G_{0,S}^<$, which is the first non-vanishing one, reads:

$$G_{0,(2)}^<(x_1,x_2) = \frac{1}{Z}\frac{n_0}{2}\int d^d y_1 d^d y_2 \langle\psi_+(x_1)\bar\psi_-(x_2)[\psi_-(\vec{y}_1,0)\bar\psi_+(\vec{y}_2,0)+\psi_-(\vec{y}_2,0)\bar\psi_+(\vec{y}_1,0)]\rangle_0$$

$$= \frac{1}{Z}\frac{n_0}{2}2e^{-\Gamma_d\frac{t_1+t_2}{2}} + \frac{Z_{(2)}}{Z}G_{0,(0)}^<(x_1,x_2). \tag{84}$$

In the first term of the second line of Eq. (84), we have again used the normalisation of the Gaussian propagator $N(\vec{x},t)$ in Eq. (55) to integrate over the spatial variables $\vec{y}_1$, $\vec{y}_2$. The last equation shows that the propagator $G_{0,S}^<$ changes compared to to the bulk propagator $G_{0,(0)}^<$ as a consequence of boundary conditions factors. One may then ask what happens to $G_{0,S}^<$ at higher orders in the expansion in $n_0$. In this simple case, as no interactions are present, all Feynman diagrams corresponding to $(2+j)$-point correlation functions ($\sim n_0^j$ with $j>1$) necessarily contain fully disconnected vacuum-to-vacuum bubbles, i.e., Feynman diagrams with no external legs. One can then factorise out the connected part, which is given by Eq. (84). The remaining multiplicative factor is given by the sum of all disconnected vacuum-to-vacuum diagrams. Hence, a recursive scheme is found, where the $2j$-th term in the power-series for $Z$ corrects the $2j+2$-th order $G_{0,(2j+2)}^<(x_1,x_2)$. Summing up all orders:

$$G_{0,S}^< = G_{0,(0)}^< + G_{0,(2)}^< + G_{0,(4)}^< + \dots = \left[n_0 e^{-\Gamma_d\frac{t_1+t_2}{2}} + G_0^<\right]\frac{1+Z_{(2)}+Z_{(4)}+\dots}{Z}$$

$$= n_0 e^{-\Gamma_d\frac{t_1+t_2}{2}} + G_0^<(x_1,x_2). \tag{85}$$

Setting $x_1 = x_2$ and remembering that $G_0^< = 0$ from (54), we recover the result of Eq. (62) for the particle density. We note, similarly to the observation made after (78), that $G_{0,S}^<$ is not time translation invariant as a consequence of the boundary terms due to the initial condition. In particular, the first term on the right hand side of (85) accounts for the dynamical approach to the vacuum stationary state ($G_0^< = 0$). The analysis of this Subsection is an example of *linked cluster theorem*, see, e.g., Refs. [4,89], which states that only the connected diagrams actually correct the correlation functions. Disconnected diagrams, instead, contain at least one

vacuum-to-vacuum bubble. These bubbles can be factorised and cancel out upon resummation with the normalisation factor $Z$. In this simple example, as the theory is non-interacting, connected diagrams appear at second order only and the propagator is determined by $G^<_{0,(2)}$. Let us eventually show that the sum of the vacuum-to-vacuum bubbles correctly retrieve the normalisation $Z = \mathcal{N}$ of the coherent state (48), $\mathcal{N} = e^{\int_x n_0}$. Discarding this term from the boundary condition (51) leads, indeed, to a different normalisation of the full generating functional, i.e., $Z_K = 1$, but $Z = \mathcal{N}$. This can be directly obtained also from the functional integral representation. In fact, it is clear that $Z_{(2)} = n_0 \int d^d y$. Calculating the full series of corrections in Eq. (81), one can check that $1 + Z_{(2)} + Z_{(4)} + ... = e^{n_0 \int d^d y} = \mathcal{N} = Z$, i.e., the sum of all vacuum-to-vacuum bubbles retrieves the normalisation of the partition function $Z$.

One can repeat the same procedure in the RAK basis (56), arriving to a similar result

$$G^K_{0,(2)}(x_1, x_2) = \frac{1}{Z} \frac{n_0}{2} 4 e^{-\Gamma_d \frac{t_1+t_2}{2}} + \frac{Z_{(2)}}{Z} G^K_{0,(0)}(x_1, x_2). \tag{86}$$

Summing up all orders of the perturbative series in $n_0$ and dividing by the partition function $Z$, the vacuum-to-vacuum bubbles cancel out and the expression in Eq. (63) is consistently recovered. Hence, evaluation at equal space-time $x_1 = x_2 = x$ yields

$$G^K_{0,S}(x, x) = 2n_0 e^{-\Gamma_d t} + i G^K_0(0), \tag{87}$$

which coincides with Eq. (64). The derivation of this section concerns the coherent initial state of Subsec. 4.1. However, the derivation can be also extended to Gibbs initial states of Subsec. 4.2. Namely the perturbative expansion of (69) generates diagrams of all order in $u_k$ (since the initial condition is in this case quadratic in the fields), which upon resummation generate the normalisation factor $\mathcal{N}_{B,F}$. The Green's functions are similarly affected only by the first order term in $u_k$ of the expansion of (69) since higher order terms contain vacuum-to-vacuum bubbles. In this way, the result (78) is eventually retrieved.

## 5 Binary annihilation

In this section we consider the case of binary annihilation $A + A \rightarrow \emptyset$ (15). Unlike single-particle decay this leads to an interacting theory. For the latter process, the Keldysh action can be readily written from Eqs. (24), (28) and (30). In the *RAK* basis it reads $[x = (\vec{x}, t)$ and the notation after (22) for space integrals]:

$$S = S_0 + S_{\text{int}} = S_0 + i\Gamma \int_{-\infty}^{\infty} dt \int_x \left[ \frac{1}{2} \bar{\phi}_c \bar{\phi}_q (\phi_c^{-\varepsilon} \phi_c^{-\varepsilon} + \phi_q^{-\varepsilon} \phi_q^{-\varepsilon}) \right. \tag{88a}$$
$$\left. - \frac{1}{2} \phi_c^\varepsilon \phi_q^\varepsilon (\bar{\phi}_c \bar{\phi}_c + \bar{\phi}_q \bar{\phi}_q) + \bar{\phi}_c \bar{\phi}_q (\phi_c^\varepsilon \phi_q^\varepsilon + \phi_c^{-\varepsilon} \phi_q^{-\varepsilon}) \right],$$

with

$$S_0 = \int_{-\infty}^{\infty} dt \int_{-\infty}^{\infty} dt' \int_x \int_{x'} (\bar{\phi}_c, \bar{\phi}_q)_x \begin{pmatrix} 0 & (G_0^A)^{-1} \\ (G_0^R)^{-1} & (G_0^{-1})_K \end{pmatrix}_{x,x'} \begin{pmatrix} \phi_c \\ \phi_q \end{pmatrix}_{x'}, \tag{88b}$$

for bosons. For the fermions, similarly, one has

$$S = S_0 + S_{\text{int}} =$$
$$= S_0 + \frac{i\Gamma}{4} \int_{-\infty}^{\infty} dt \int_x \left[ (-\vec{\nabla}\bar{\phi}_1\bar{\phi}_1 + \vec{\nabla}\bar{\phi}_1\bar{\phi}_2 + \vec{\nabla}\bar{\phi}_2\bar{\phi}_1 - \vec{\nabla}\bar{\phi}_2\bar{\phi}_2) \cdot (\phi_1^\varepsilon \vec{\nabla}\phi_2^\varepsilon + \phi_2^\varepsilon \vec{\nabla}\phi_1^\varepsilon) \right.$$
$$\left. + (\vec{\nabla}\bar{\phi}_1\bar{\phi}_2 + \vec{\nabla}\bar{\phi}_2\bar{\phi}_1) \cdot (\phi_1^{-\varepsilon} \vec{\nabla}\phi_1^{-\varepsilon} + \phi_1^{-\varepsilon} \vec{\nabla}\phi_2^{-\varepsilon} + \phi_2^{-\varepsilon} \vec{\nabla}\phi_1^{-\varepsilon} + \phi_2^{-\varepsilon} \vec{\nabla}\phi_2^{-\varepsilon}) \right], \tag{89a}$$

with the quadratic part $S_0$ as

$$S_0 = \int_{-\infty}^{\infty} dt \int_{-\infty}^{\infty} dt' \int_x \int_{x'} (\bar{\phi}_1, \bar{\phi}_2)_x \begin{pmatrix} (G_0^R)^{-1} & (G_0^{-1})_K \\ 0 & (G_0^A)^{-1} \end{pmatrix}_{x,x'} \begin{pmatrix} \phi_1 \\ \phi_2 \end{pmatrix}_{x'}. \tag{89b}$$

In Eqs. (88a) and (89a), we used the notation $\phi^{\pm\varepsilon} = \phi(t \pm \varepsilon)$ in the interaction vertices for the time shift regularisation, which will be needed to evaluate tadpole diagrams. The coupling constant $i\Gamma$, appearing in the Keldysh actions in Eqs. (88a) and (89a), is purely imaginary, due to the interactions being entirely originated by the dissipation in the Lindblad dynamics. In both the bosonic and fermionic cases, interactions vertices are quartic in the fields. This is in contrast to classical binary annihilation, where also cubic interaction vertices are present, cf. Appendix A. For fermions, in addition, spatial gradients of the fields are present.

In both Eqs. (88b) and (89b), the bare inverse propagators are defined as

$$(G_0^{R/A})^{-1}(x, x') = \frac{1}{\hbar}(i\hbar\partial_t + J\nabla_x^2 - V(\vec{x}) \pm i0^+)\delta(x - x'), \tag{90a}$$

$$(G_0^{-1})_K(x, x') = 2i0^+ F_0(x, x'). \tag{90b}$$

Here, we have further introduced the infinitesimal shifts $\pm i0^+$, which account for the retarded and advanced nature of the propagators. In Sec. 4, this regularisation was not needed since a finite one-body decay $\Gamma_d \neq 0$ already shifts the poles in the lower [upper] half of the complex plane for $G^R(\epsilon)$ [$G^A(\epsilon)$].

We note that in Eqs. (88a)-(89b) all time integrals are evaluated on the whole real axis $t \in (-\infty, \infty)$. At this point one may then wonder how the initial boundary terms are kept into account. Indeed, as discussed in Sec. 4, these initial boundary terms restrict the time integration axis in $t \in (0, \infty)$. The initial conditions of Subsecs. 4.1 and 4.2 are, however, stationary for the Hamiltonian dynamics. This means that the bare Green's functions $G_0(\vec{x}_1, \vec{x}_2, t_1 - t_2)$ associated to $S_0$ are functions of $t_1 - t_2$ only. These Green's function can be equivalently obtained by Gaussian inversion of $S_0$ in (88b) and (89b). In this case, the information on the initial occupation function $F_0$ is enforced by introducing a regularisation factor $2i0^+ F_0$ (90), with $F_0$ the initial distribution function (39), in the $q-q$ and $1,2$ components for bosons and fermions, respectively [65, 90]. This results in the very same Green's functions ($\Gamma_d = 0$ and $V(\vec{x}) = 0$) we obtained in Eqs. (63) and (78). In this way, both the initial conditions discussed in Sec. 4 are implemented in a quadratic bare action $S_0$. The interaction part (88a) and (89a) in $\Gamma$ can then be treated by perturbative expansion with respect to $S_0$ via Wick's theorem. This is the approach we will follow in the next Subsections and this is why we reported in Eqs. (88a)-(89b) all the time integrals over the whole real line. In any case, at the level of the kinetic equation, the regularisation factor $2i0^+ F$ becomes soon unimportant since a finite value of $(G^{-1})_K$ is produced by the interactions. The information on the initial occupation function is then eventually enforced as an initial condition to the obtained kinetic equation.

In Subsec. 5.1, we show that the interaction terms in the Keldysh action do not modify the normalisation of the Keldysh partition function. In Subsec. 5.2, we set up the study of $A+A \rightarrow \emptyset$ via kinetic equations of the Boltzmann form for the particle density. We show how these equation naturally emerges in the Euler-scaling limit of weak dissipation. In Subsecs. 5.3 and 5.4, we eventually specialise the derivation of the kinetic equations to bosons and fermions, respectively. The convention used to depict Feynman diagrams is the one defined in Fig. 2.

## 5.1 Normalisation of the Keldysh partition function

We explicitly compute first-order perturbative corrections in $\Gamma$ to $Z_K$, and demonstrate that their contribution is identically vanishing, i.e., that $Z_{K,(1)} = 0$. This is done for the bosonic case, but can be easily extended to the fermionic one.

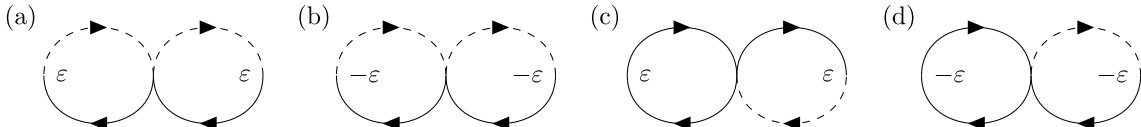

Figure 3: **Diagrammatic representation of** $Z_{K,(1)}$. Diagrams constituting the first-order perturbative corrections to the Keldysh partition function $Z_K$. Feynman rules for propagators are given in Fig. 2, i.e., solid lines here represent classical fields, while dashed lines are quantum fields. The $\varepsilon$ in the diagram corresponds to the time argument of the associated diagram. In (a), for example, we have $G^A(\varepsilon)G^R(\varepsilon)$. Because of the regularisation of the dissipative interaction vertices introduced in Subsec. 3.3, all diagrams vanish, i.e., $Z_{K,(1)} = 0$. By the same token, higher-order corrections $Z_{K,(m)}$ with $m > 1$ also vanish.

Taylor-expanding $\exp(iS_{\text{int}})$ in powers of $\Gamma$ as in Eq. (81) and averaging $\langle\ldots\rangle_0$ over the gaussian weight $\exp(iS_0)$ one obtains a power-series of free $4m$-point Green's functions of order $\Gamma^m$. Let us truncate the expansion at first order in $\Gamma$, i.e., considering 4-point Green's functions. The diagrams are obtained by contracting the fields in each vertex via Wick's theorem. As no external legs are present, the diagrams are pictorially represented by vacuum-to-vacuum bubbles, similarly to Subsec. 4.3 where vacuum-to-vacuum diagrams in the expansion of $Z$ in the initial boundary conditions were considered. The diagrams where vanishing $qq$-propagators are present are zero. We then remain with the following expression for $Z_{K,(1)}$, whose diagrammatic representation is given in Fig. 3:

$$Z_{K,(1)} = i\Gamma \int_{x,t} \left[ 2G^A_\varepsilon G^R_\varepsilon + 2G^A_{-\varepsilon} G^R_{-\varepsilon} + \frac{3}{2} G^K_{-\varepsilon} G^R_{-\varepsilon} - \frac{3}{2} G^A_\varepsilon G^K_\varepsilon \right]. \tag{91}$$

In this equation, we denote with $G^{\mu\nu}_{\pm\varepsilon} = G^{\mu\nu}_0(t+\varepsilon/2, t-\varepsilon/2) = G^{\mu\nu}_0(\varepsilon)$ (given in Eq. (57) with $\Gamma_d = 0$) the Green's functions evaluated according to the time regularisation of the interaction vertices (88a). Note that it is fundamental to keep track of this regularisation since vacuum-to-vacuum bubbles as in Fig. 3 involve contractions of fields at the same space-time vertex and therefore lead to the ambiguity of fixing $\Theta(0)$. Clearly, as $G^{R/A}(t_1, t_2) \sim \Theta(\pm(t_1 - t_2))$, each term contains at least a null propagator $G^R_{-\varepsilon} \sim \Theta(-\varepsilon) = 0$ and $G^A_\varepsilon \sim \Theta(\varepsilon) = 0$, entailing that $Z_{K,(1)}$ is identically vanishing. This procedure can be hierarchically extended to higher-$m$ order corrections $Z_{K,(m)}$, implying that the Keldysh partition function $Z_K = 1$ is not modified by the presence of dissipative interactions. This calculation is directly extended to the fermionic case, where Wick's theorem can directly applied to the set of 4-point interacting Green's functions. The difference is that interaction vertices must now be considered with their spatial derivatives as defined in Eq. (89a). We will show in Subsec. 5.4 how to compute the related expectation values. Nevertheless, one can show that, again, all vacuum-to-vacuum bubbles in the interaction vertices vanish because of the regularisation prescription.

## 5.2 Boltzmann equation from the Euler-scaling limit of the self-energy

We aim to study the system in the weak dissipation regime $\Gamma \to 0$. This is the reaction-limited regime. In particular we consider the *Euler-scaling limit* [69, 70], where the space and time are simultaneously sent to infinity and their ratio is kept finite:

$$x, t \to \infty, \quad \Gamma \to 0, \quad \text{with} \quad \bar{x} = \Gamma x, \quad \bar{t} = \Gamma t \quad \text{fixed.} \tag{92}$$

We are hence interested in the large-scale properties of the system, i.e., on the dynamics taking place on large space-time scales $x, t \sim \Gamma^{-1}$. Simultaneously, we consider the regime where the

external potential $V(\vec{x})$ in (88a) and (89a) is slowly varying on a length scale set by $\Gamma$:

$$V(\bar{\vec{x}}) = V(\Gamma\vec{x}). \tag{93}$$

The reaction-limited regime can be therefore identified with a weak interaction regime, in the sense that the coupling $i\Gamma$ to the quartic parts of the action (88a) and (89a) is small. In this regime, the system can be still locally described by the density of quasiparticles in phase space. The interaction scrambles the density of quasiparticles (44) and induce a finite lifetime. The large-scale equation describing these phenomena is the Boltzmann equation, which we here obtain from the Euler-scaling limit (92) of the expansion in $\Gamma$ of the Keldysh action (88a) and (89a).

The starting point of the analysis is the Dyson equation for the dressed Green's functions $\hat{G}$:

$$[\hat{G}_0^{-1} - \hat{\Sigma}] \circ \hat{G} = \mathbb{1}, \tag{94}$$

where $\circ$ notation refers to the convolution product according to the definition (40). In the previous equation, $\hat{\Sigma}$ is called *self-energy* matrix and it describes how interactions modify the bare propagator $\hat{G}_0$ turning it into the dressed one $\hat{G}$. The self-energy matrix is obtained by summing all the irreducible diagrams [64–66], namely those diagrams which cannot be subdivided into two disconnected diagrams by cutting a single internal propagator line. Irreducible diagrams feature an internal sector, displaying loops, and two external propagators. Depending on the two external propagators, $\hat{\Sigma}$ can be written as a 2x2 matrix, whose 2-valued indices take value depending on the external propagator jointed on each side. The self-energy matrix for bosons $\hat{\Sigma}_B$ and fermions $\hat{\Sigma}_F$ reads as

$$\hat{\Sigma}_{\mathrm{B}}(y_1, y_2) = \begin{pmatrix} 0 & \Sigma^A(y_1, y_2) \\ \Sigma^R(y_1, y_2) & \Sigma^K(y_1, y_2) \end{pmatrix}, \tag{95a}$$

$$\hat{\Sigma}_{\mathrm{F}}(y_1, y_2) = \begin{pmatrix} \Sigma^R(y_1, y_2) & \Sigma^K(y_1, y_2) \\ 0 & \Sigma^A(y_1, y_2) \end{pmatrix}. \tag{95b}$$

The different structure of $\hat{\Sigma}$ in the two cases follows from the different definition of Keldysh rotation (27) and (29). For bosons, specifically, $\hat{\Sigma}_B$ in the Keldysh indices has the same structure as the inverse $\hat{G}_B^{-1}$ of the propagators matrix (34). For fermions, instead, $\hat{\Sigma}_F$ has the same structure as the matrix $\hat{G}_F$ (35) itself. Points $y_1$, $y_2$ are the space-time coordinates of the interaction vertices where $\Sigma$ connects to the external legs, and as such are internal vertices which must be integrated out. Interestingly, the causality structure of the matrix of inverse Green's functions ensued by probability conservation in Keldysh theory, can be extended to the self-energy as well. Accordingly, its classical-classical (2,1 entry for fermions) entry identically vanishes. From the Dyson equation the Keldysh entry Eq. (94) yields the coupled equation:

$$[(G_0^{-1})^R - \Sigma^R] \circ G^K = \Sigma^K \circ G^A. \tag{96}$$

We note that, due to the interactions, $\Sigma^K$ acquires a nonvanishing finite value and therefore $(G^{-1})_K$ similarly becomes finite. This is the reason why we can drop the regularatisation factor $2i0^+F_0$ in the derivation of the Boltzmann equation, as hinted after Eq. (89b). The previous equation carries information on occupation density of the system. In order to make such information more explicit, we use the parameterisation (39) of the Keldysh Green's function in terms of the distribution function $F$. One then writes a *quantum kinetic equation* for the distribution function $F(y_1, y_2)$:

$$\left[ -i(\partial_{t_1} + \partial_{t_2}) - J(\nabla_{\vec{x}_1}^2 - \nabla_{\vec{x}_2}^2) + (V(\vec{x}_1) - V(\vec{x}_2)) \right] F(x_1, x_2) = \tilde{I}_{\mathrm{coll}}[F]. \tag{97}$$

The expression on the left hand side is called the kinetic term, while we refer henceforth to the right hand side as the collision integral. The latter is written as

$$\tilde{I}_{\text{coll}}[F] = \Sigma^K \circ \mathbb{1} - (\Sigma^R \circ F - F \circ \Sigma^A). \tag{98}$$

Equations (97) and (98) completely describe the microscopic dynamics of the model. In order to derive an effective description for the large-scale slow degrees of freedom we now take the Euler-scaling limit (92) and (93). In the present context this can be achieved exploiting the Wigner transform (43). In the Wigner coordinates (42), the Euler scaling limit reads as

$$x_1, x_2 \to \infty, \ \Gamma \to 0, \quad \text{with} \quad \bar{x} = \Gamma \frac{x_1 + x_2}{2} \quad \text{fixed}, \quad \Gamma(x_1 - x_2) \ll 1. \tag{99}$$

This limit identifies the centre of mass as the the slow variable, which changes on a large scale $x \sim \Gamma^{-1}$. Vice versa, the relative coordinate $x' = x_1 - x_2$ changes on a much shorter scale. The fast dependence on $x'$ can then be integrated out by Wigner transform deriving an effective equation for the slow-emergent degree of freedom. This approach is routinely followed in deriving kinetic equations from Keldysh formalism, see, e.g., Refs. [64–67]. In particular, within the limit (99), one may turn the convolution $C(x_1, x_3)$ of any two space-time two-point functions $A(x_1, x_3)$, $B(x_3, x_2)$ into a derivative expansion of the respective Wigner transforms $A(x, k)$, $B(x, k)$, akin to the Moyal star-product expansion performed in hydrodynamics [91–96]. In particular, in the Euler scaling limit, each derivative comes with a scaling factor $\Gamma$ and therefore one can truncate the series of phase-space derivatives with its zeroth [first] order, namely to the term corresponding to the product of their transforms [of the first derivatives of their transforms]:

$$C(x_1, x_2) = A(x_1, x_3) \circ B(x_3, x_2) \xrightarrow{\text{WT}} C(x, k) = AB + \frac{i\Gamma}{2}[\partial_{\bar{x}} A \partial_k B - \partial_k A \partial_{\bar{x}} B] + \mathcal{O}(\Gamma^2). \tag{100}$$

We call the truncation of the derivative expansion to the first order the Wigner approximation. The expression (100) singles out the slow modes, whose dynamics take place on the long space-time scales $\sim \Gamma^{-1}$. The fast dynamics occurring on shorter space-time scales, which is contained in the higher orders of the derivative expansion, is zoomed out in the Euler limit (99). Applying the prescription (100) to Eqs. (97) and (98) one obtains

$$i\Gamma\Big[\partial_{\bar{t}} + \vec{v}_g(\vec{k})\cdot\vec{\nabla}_{\bar{x}} - \frac{1}{\hbar}\vec{\nabla}_{\bar{x}}V(\vec{\bar{x}})\cdot\vec{\nabla}_k\Big]F(\bar{x},k) = \tilde{I}_{\text{coll}}[F] = i\,\Sigma^K(x,k) + 2F(x,k)\text{Im}\Sigma^R(x,k). \tag{101}$$

An important point is here in order. Namely in (101) we assumed that $\text{Re}\Sigma^R(x, k) = 0$. The real part of the self energy, indeed, generically contributes to dressing $\epsilon_k(\vec{x}) = Jk^2 + V(\vec{x}) + \text{Re}\Sigma^R(x, k)$ the quasi-particle dispersion relation $\epsilon_k(\vec{x})$ as a consequence of interactions. In the present case, where interaction terms are purely dissipative, we will show in Subsecs. 5.3 and 5.4 that this assumption holds true. Namely, in the Euler scaling limit, the self energy is purely imaginary – $\text{Re}\Sigma^R = 0$ – and therefore the dispersion relation simply remains $\epsilon_k(\vec{x}) = Jk^2 + V(\vec{x})$. This expression is the one obtained within the local density approximation, which is valid for a slowly varying potential as in Eq. (93) (cf. also Appendix B for additional details). The associated group velocity $v_g(\vec{k}) = \nabla_k \epsilon_k(\vec{x})/\hbar = 2Jk/\hbar$. The imaginary part of the self energy $\text{Im}\Sigma^R$ appears, instead, inside the collision integral and it determines the finite quasi-particle lifetime due to dissipation. We note that the left hand side of this equation is proportional to $\Gamma$. Consequently, in order to have a finite scaling limit according to Eq. (99), one needs to consider terms of $\tilde{I}[F]$ which are linear in $\Gamma$. At the Euler-scale, therefore, the only diagrams which determine the Boltzmann equation come from first-order terms in the self energy. These diagrams, as we detail in Subsecs. 5.3 and 5.4, are of tadpole form.

In order to bring (101) to a calculable form for the phase-space density (44), one needs to rely on one additional assumption. In particular, the phase-space density $n(\vec{x}, t, \vec{k})$ provides the quasi-particle basis [70, 97] for the representation of conserved charges of the Hamiltonian (12). We consequently expect $n(\vec{x}, t, \vec{k})$ to be the emergent slow degree of freedom as long as quasi-particles are well defined. This is true in the limit where the spectral function $A(x, k) \sim \delta(\epsilon - \epsilon_k(\vec{x}))$ is a sharply peaked function of $\epsilon$. In the weak dissipation regime, as we explain in Appendix B, this is still true and we can therefore introduce the so-called "on-shell" distribution function

$$\tilde{F}(\vec{x}, t, \vec{k}) \equiv F(\vec{x}, t, \vec{k}, \epsilon = \epsilon_k(\vec{x})). \tag{102}$$

We note that $\tilde{F}$ turns into a function of only three variables. The mass-shell approximation of the distribution function $\tilde{F}$ therefore amounts to saying that quasiparticles remain well-defined even throughout time evolution, albeit with a finite, but long, lifetime. This is precisely the reason allowing to recast (101) in the form of a Boltzmann equation for the occupation function $n(\vec{x}, t, \vec{k})$. The latter is connected to the on-shell distribution function as follows:

$$\tilde{F}(\vec{x}, t, \vec{k}) \approx 2\pi \int \frac{d\epsilon}{2\pi} F(x, k) \delta(\epsilon - \epsilon_k(\vec{x}))$$
$$= \int \frac{d\epsilon}{2\pi} iG^K(x, k) = iG^K(\vec{x}, t, \vec{k}, t' = 0) = 1 + 2\xi n(\vec{x}, t, \vec{k}). \tag{103}$$

The first equality in Eq. (103) relies on the on-shell assumption and therefore on the peaked structure $A(x, k) \sim \delta(\epsilon - \epsilon_k(\vec{x}))$ of the spectral function. The second equalities follows from the parameterisation (39) and its Wigner transform in the Euler-scaling. The third equality is the definition of the inverse Wigner transform, which leads to the Keldysh Green's function $G^K(\vec{x}, t, \vec{k}, t' = 0)$ evaluated at equal times $t' = 0$. In the last equality, we used (44). We note that the quasi-particle assumption (102) is the key assumption necessary to connect the Keldysh kinetic approach of the manuscript to the TGGE approach of Refs. [36, 38, 40, 42–44, 54, 57]. The TGGE method relies, indeed, on the existence of stable excitations, the quasiparticles, which are labelled by the momentum $k$. These excitations are stable as a consequence of the entirely elastic scattering they undergo [98]. Elastic scattering is, in turn, determined by the extensive number of conservation laws associated to the Hamiltonian (12). In the presence of weak integrability breaking interaction (in our case dissipation), quasi-particles are no longer stable, but they can still be defined since they decay on a long time scale. In the Keldysh language, the statement of well-defined quasi-particles is precisely given by (102), which can be therefore interpreted as the existence of a GGE state describing the system dynamics. The time dependence of the GGE, hence the name TGGE, follows from the collision integral, i.e., the slow decay of the quasi-particles and conserved charges expectation values.

## 5.3 The bosonic Boltzmann equation

In the Euler-scaling limit (99), the collision integral $\tilde{I}[F]$ is determined by terms at first order in $\Gamma$. Diagrams of order $\Gamma$ associated to the interaction vertices Eq. (88a) are tadpole and they are drawn in Fig. 4. These diagrams determine the self-energy $\Sigma_{(1)}^{R/A/K}$ entries as

$$\Sigma_{(1)}^R(y, y') = \Gamma\delta(y')[G_{0,-\varepsilon}^A(y, y') + G_0^K(y, y')], \tag{104a}$$

$$\Sigma_{(1)}^A(y, y') = \Gamma\delta(y')[G_{0,\varepsilon}^R(y, y') - G_0^K(y, y')], \tag{104b}$$

$$\Sigma_{(1)}^K(y, y') = \Sigma_{(1)}^R(y, y') - \Sigma_{(1)}^A(y, y'), \tag{104c}$$

where we employed, following the equal-time regularisation prescription of (3.3), cf. also Fig. 3, the regularised retarded (advanced) Green's functions $G_{0,\varepsilon}^R$ ($G_{0,-\varepsilon}^A$)

$$G_{0,\varepsilon}^R(\vec{y}, t, \vec{y}', t') = -iN(\vec{y}', t')e^{-iV(\Gamma\vec{y})t'}, \quad \text{and} \quad G_{0,-\varepsilon}^A(\vec{y}, t, \vec{y}', t') = iN(\vec{y}', t')e^{-iV(\Gamma\vec{y})t'}. \tag{105}$$

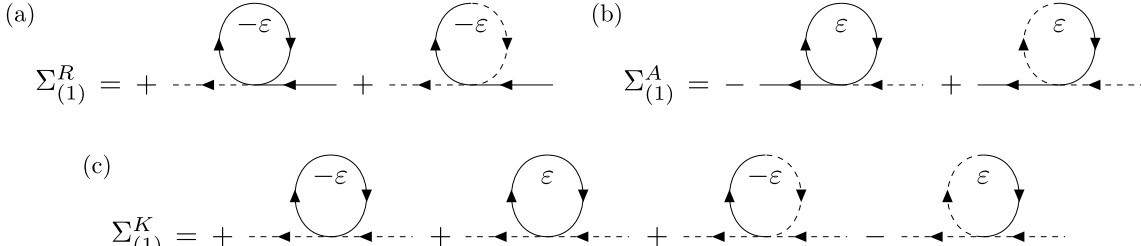

Figure 4: **Self-energy diagrams for the bosonic binary annihilation.** Feynman diagrams for the (a) retarded $\Sigma^R_{(1)}$, (b) advanced $\Sigma^A_{(1)}$ and (c) Keldysh $\Sigma^K_{(1)}$ entries of the self-energy matrix $\hat{\Sigma}$ at first order in perturbation theory in $\Gamma$. The diagrammatic conventions are the same as in Fig. 2. Even though internal loops exclusively contribute to the self-energy, the external legs are also drawn in figure, in order to clarify the meaning of the $c/q$-valued Keldysh matrix indices. For example, in $\Sigma^R_{(1)}$, one has a dashed line exiting the internal vertex ($\bar{\phi}_q$) and one solid line entering it ($\phi_c$) and it is therefore associated to the retarded $c - q$ entry. Each diagram has an implicit factor $\Gamma$ in front, while the relative signs are reported in the figure.

Furthermore, we use the Wigner variables (42) $y = (y_1 + y_2)/2$ and $y' = y_1 - y_2$ for the centre of mass and the relative coordinate, respectively. The expressions (105) are computed from contractions as $G^R_{0,\varepsilon} = -i\langle\phi^\varepsilon_c \bar{\phi}_q\rangle_0$ and $G^R_{0,\varepsilon} = -i\langle\phi_c \bar{\phi}^{-\varepsilon}_q\rangle_0$ [$G^A_{0,\varepsilon} = -i\langle\phi^{-\varepsilon}_q \bar{\phi}_c\rangle_0$ and $G^A_{0,\varepsilon} = -i\langle\phi_q \bar{\phi}^\varepsilon_c\rangle_0$] in (88a) tracking the time regularisation shift $\varepsilon$. Because of the same regularisation, contractions of vertices generating $G^R_{0,-\varepsilon} = G^A_{0,\varepsilon} \sim \Theta(-\varepsilon) = 0$ are identically zero. The diagrams associated to the self energy in Eq. (104) are drawn in Fig. 4. The self-energy $\hat{\Sigma}_{(1)}$ of Eq. (104) clearly displays the causality structure which is also typical of the matrix $\hat{G}$ of Green's functions. The classical-classical entry is indeed identically vanishing, while one has that $[\Sigma^R_{(1)}]^\dagger = \Sigma^A_{(1)}$ and $[\Sigma^K_{(1)}]^\dagger = -\Sigma^K_{(1)}$, using Eqs. (38).

The expressions appearing in Eq. (104) are the bare free Green's functions $G^R_0$ and $G^A_0$ associated to the quadratic action (88b) in regime where the potential $V$ is slowly varying according to Eq. (93). The expressions are explicitly reported in App. B [cf. Eq. (B.5)]. For the Keldysh Green's function $G^K_0$, we consider the parameterisation $G^K_0 = G^R_0 \circ F - F \circ G^A_0$ and keep $F$ as the unknown of the quantum kinetic equation, in order for it to be self-consistently derived. This approach is referred to as perturbative Born approximation. One can also evaluate the self energy (104) in terms of the dressed Green's functions $G^{R,A}$, which should then be determined self-consistently [99]. The latter approach is called self-consistent Born approximation and it yields non-perturbative results since it amounts to resumming the infinite class of one-particle reducible diagrams obtained by concatenating the tadpole structure of Fig. 4. In this manuscript, we do not use it because this resummation leads to terms of order $\mathcal{O}(\Gamma^m)$, with $m > 1$, which are subleading in the scaling limit (99).

We now proceed to the evaluation of the collision integral $\tilde{I}[F]$ for bosonic binary annihilation. We first need to Wigner-transform the self-energy entries in Eq. (104). This can be done by exploiting the inverse of the Wigner convolution theorem (100), which transforms products of two-point functions into convolutions of their Wigner transforms [65]. This leads to the following self-energy terms:

$$\Sigma^{R/A}(x,q) = \Gamma \int \frac{d^d k}{(2\pi)^d}\frac{d\epsilon}{2\pi}\left[\pm G^K_0(x,k) + G^{R/A}_{0,\varepsilon}(x,q)\right], \qquad (106)$$

$$\Sigma^K(x,q) = \Sigma^R(x,q) - \Sigma^A(x,q) = \Gamma \int \frac{d^d k}{(2\pi)^d}\frac{d\epsilon}{2\pi}\left[2G^K_0(x,k) - 2i\text{Im}G^R_{0,\varepsilon}(x,k)\right], \qquad (107)$$

with the Wigner transform of the regularised retarded and advanced Green's function (105)

$$G_{0,\varepsilon}^{R/A}(x,q) = \mp 2\pi i \delta(\epsilon - \epsilon_k(\vec{x})). \tag{108}$$

From the conjugation properties in Eq. (38), it follows that $G_0^K(x,k)$ is purely imaginary. It is then clear that the self-energy matrix elements in these expressions are purely imaginary quantities: $\mathrm{Re}\Sigma^R(x,q) = 0$. This is the result we anticipated after Eq. (101) and it implies that the dispersion relation $\epsilon_k(\vec{x})$ and the external potential $V$ are not renormalised by dissipative interactions in the Euler scaling limit, as it is, instead, generically the case for Hamiltonian interactions. In addition, the self energy entries $\Sigma^{R,A,K}(x,q)$ are, in this case, actually independent on $q$. After rearranging the terms as in Eq. (101), and writing $G_0^K$ in terms of $F$ in order to exploit the self-consistent approach, the collision integral reads:

$$I[F] = \Gamma \int \frac{d^d k}{2\pi} \int \frac{d\epsilon}{2\pi} \big[1 - F(k) - F(q) + F(k)F(q)\big] 2\mathrm{Im}G_{0,\varepsilon}^R(x,k). \tag{109}$$

We can now remove the integral over frequencies $\epsilon$ using Eq. (108), thus considering the on-shell distribution $\tilde{F}$ according to the quasi-particle approximation (102). Enforcing the relation between the on-shell distribution function and the occupation function $n(x,\vec{k})$ given in Eq. (103), we eventually obtain the kinetic equation in terms of $n(x,\vec{k})$. Reintroducing the notation with separated rescaled space $\bar{x} = \Gamma\vec{x}$ and time $\bar{t} = \Gamma t$ variables, Eq. (99), this has the shape of the following Boltzmann-like equation:

$$\left[\partial_{\bar{t}} + \vec{v}_g(\vec{k}) \cdot \vec{\nabla}_{\bar{x}} - \frac{1}{\hbar}\vec{\nabla}_{\bar{x}}V \cdot \vec{\nabla}_k\right] n(\vec{\bar{x}},\bar{t},\vec{k}) = -4 \int \frac{d^d q}{(2\pi)^d} n(\vec{\bar{x}},\bar{t},\vec{k}) n(\vec{\bar{x}},\bar{t},\vec{q}). \tag{110}$$

The kinetic term describes the ballistic spreading of quasiparticles, with group velocity $\vec{v}_g(\vec{k}) = \hbar\vec{k}/m$, under the drive of an external force $-\vec{\nabla}_x V/\hbar$. The collision integral characterises rearrangements of the momentum occupation function $n(\vec{\bar{x}},\bar{t},\vec{k})$ due to interactions. Because the interaction here considered is purely dissipative, the right hand side of Eq. (110) describes particle losses. As a consequence of this, the collision integral is strictly negative $I[n] < 0$ implying that the density of particles is strictly descreasing. We also notice that the decay of a mode $n(\vec{\bar{x}},\bar{t},\vec{k})$ does not depend on the value of the mode $\vec{k}$ itself. Modes initially equally occupied decay therefore at the same rate.

The Boltzmann equation (110) is valid in arbitrary dimension $d$. In the homogeneous case, i.e., when the initial state is translational invariant and no potential is present $V = 0$, the Wigner function reduces to the occupation function in momentum space $n(\vec{\bar{x}},\bar{t},\vec{k}) = n(\bar{t},\vec{k})$. The integral over $\vec{k}$ of Eq. (110) leads to a closed equation for the spatial density:

$$\frac{dn(\bar{t})}{d\bar{t}} = -4n^2(\bar{t}), \quad \text{with} \quad n(\bar{t}) = \int \frac{dk^d}{(2\pi)^d} n(\bar{t},\vec{k}), \tag{111}$$

which coincides with the homogeneous classical rate equation for pair annihilation (see also the discussion in Appendix A). For binary annihilation in the noninteracting Bose gas the density decays according to the mean field exponent (1) in arbitrary spatial dimensions $d$:

$$n(\bar{t}) = \frac{n_0}{1 + 4n_0\bar{t}} \sim \bar{t}^{-1}. \tag{112}$$

In one spatial dimension $d = 1$, the Boltzmann equation here above matches previous predictions for the Bose gas subject to weak binary losses derived within the TGGE framework [36,57]. In Ref. [36], in addition, the Bose gas is studied also in the presence of Lieb-Liniger quartic interactions. In that case, in Eq. (110) not only the collision integral is modified by the

interaction, but also the kinetic term. The group velocity is, in particular, dressed by the interaction. This effect, within the Keldysh formalism is expected to emerge from nonvanishing real parts of the self-energy $\Sigma^R$ due to the Hamiltonian interactions. It is, however, hard to explicitly show this aspect since one would need to resum diagrams of all order in the Hamiltonian interaction. The result of [36] is, indeed, based on the generalised hydrodynamics [100,101] description of the interacting Bose gas and, as such, it is nonperturbative in the Hamiltonian interaction. We further discuss on the relation between the TGGE ansatz and our perturbative expansion in the next Subsec. 5.4 for fermions.

## 5.4 The fermionic Boltzmann equation

The derivation of the Boltzmann equation for fermions requires some additional care compared to the case of bosons. From the technical point of view, indeed, the nearest neighbour annihilation (9) arising from the fermionic repulsion introduces additional spatial gradients (15) in the interaction vertices (89a). These gradients render the self-energy $\hat{\Sigma}$ a differential operator acting from both sides on the free propagators. In the Euler-scaling limit, we again consider only first-order terms in $\Gamma$, which have the form of tadpole diagrams. Hence, whenever two fields from an internal loop or from an external leg are contracted together with 0, 1, 2 differential operators, the corresponding propagator is differentiated 0, 1, 2 times, namely one has that:

$$\langle \vec{\nabla}\phi_\mu(x_1)\vec{\nabla}\bar{\phi}_\nu(x_2)\rangle_0 = \lim_{a\to x_1}\lim_{b\to x_2}\vec{\nabla}_a\vec{\nabla}_b G_0^{\mu\nu}(a,b). \tag{113}$$

We recall that fields $\phi_\mu$, with $\mu = 1,2$, are entering the vertices, whereas conjugated fields $\bar{\phi}_\mu$ are outgoing from those, so that the association between diagrams and propagators still follows the convention used for bosonic fields, i.e., the representation given in Fig. 2. Moreover, we use blue lines to indicate spatial gradients of the corresponding fields according to (113). The tadpole diagrams associated to the fermionic self-energy at order $\Gamma$ are reported in Fig. 5.

The derivation of the Boltzmann equation from the diagrams in Fig. 5 then proceeds along similar lines as in the case of bosons. The details have been worked out in Ref. [56] and we therefore do not report them here for the sake of brevity. The resulting Boltzmann-like equation in the Euler scaled variables $\bar{\vec{x}} = \Gamma\vec{x}$ and $\bar{t} = \Gamma t$ reads:

$$\left[\partial_{\bar{t}} + \vec{v}_g(\vec{k})\cdot\vec{\nabla}_{\bar{x}} - \frac{1}{\hbar}\vec{\nabla}_{\bar{x}}V\cdot\vec{\nabla}_k\right]n(\bar{\vec{x}},\vec{k},\bar{t}) = -\int\frac{d^dq}{(2\pi)^d}(\vec{k}-\vec{q})^2\,n(\bar{\vec{x}},\vec{k},\bar{t})\,n(\bar{\vec{x}},\vec{q},\bar{t}). \tag{114}$$

We mention that the collision integral contains in principle also terms proportional to the spatial gradient $\nabla_{\bar{x}}n$ of the Wigner function. We neglect these terms here, as they are negligible in the Euler scaling limit according to Eq. (100). The action of $\hat{\Sigma}$ as a differential operator on real functions, as in the right hand side of Eq. (98), yields purely imaginary functions. As a consequence, as in the case of bosons, no renormalisation of the quasi-particle dispersion relation and potential. The factor $(\vec{k}-\vec{q})^2$ is the main difference with respect to the Boltzmann equation (110) for bosons. It includes the effect of the fermionic anticommutative statistics, entailing nearest-neighbour interaction, which is absent in the bosonic theory. This difference in the collision integral yields a largely enriched dynamics of the Fermi gas, which displays an interesting behaviour in both homogeneous and non-homogeneous scenarios.

In the homogeneous case, in particular, a closed rate equation for the particle density $n(\bar{t})$ as in Eq. (111) cannot be obtained. The density accordingly does not follow the mean field prediction and it decays in generic dimensions $d$ as [56]

$$n(\bar{t}) \sim \bar{t}^{-\frac{d}{d+1}}. \tag{115}$$

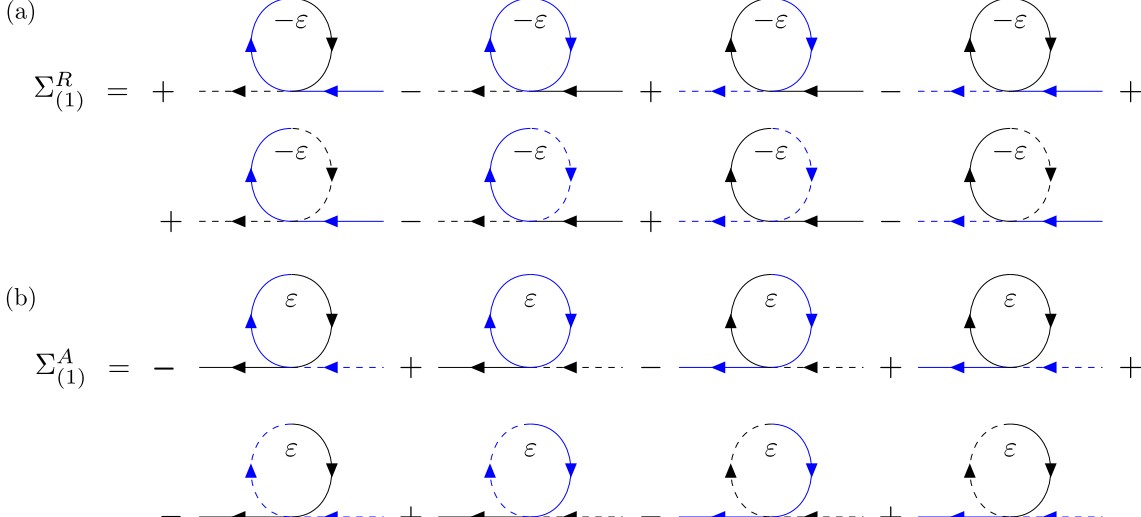

Figure 5: **Self-energy diagrams for the fermionic binary annihilation.** Feynman diagrams for the (a) retarded $\Sigma^R_{(1)}$ and (b) advanced $\Sigma^A_{(1)}$ entries of the self-energy matrix at first order $\Gamma$. The Keldysh component is given by $\Sigma^K_{(1)} = \Sigma^R_{(1)} - \Sigma^A_{(1)}$ and its diagrammatic representation is not reported for the sake of brevity. External legs are drawn in the figure in order to make explicit the meaning of the 1/2-valued Keldysh matrix indices in Eq. (95b), and how derivatives affect the external propagators. Namely, an external blue left leg denotes differentiation with respect to the second argument of the ensuing propagator, while an external blue right leg denotes differentiation with respect to the first argument of the ensuing propagator. All the other diagrammatics conventions are as in Fig. 2. Each diagram has an implicit factor $\Gamma/4$, while the relative sign is reported.

This result should be contrasted with the analogous result (112) for homogeneous bosons. Here, mean field decay is valid in any $d$, while in the fermionic case, on the contrary, deviations from mean field are present in any dimension $d$. For fermions the mean-field decay is approached only asymptotically for large $d$ values. In inhomogeneous cases as well, as we discuss in detail in the next Sec. 6, the collision integral also determines rich dynamics for the Wigner function $n(\vec{x}, \vec{k}, \bar{t})$. Differently from the bosonic case, the modes $\vec{k}$ do not decay all at the same speed and one therefore obtains non-trivial profiles for the Wigner function in phase space $(\vec{x}, \vec{k})$.

It is also important to mention that in spatial dimension $d = 1$, Eq. (114) coincides with previous results derived assuming the TGGE relaxation ansatz in Refs. [38, 42–44]. The result of the analysis is a Boltzmann-like equation akin to (114), where the collision integral again describes losses on the lattice. The continuum space limit of these results has been carried out in Ref. [40] and it leads to (114) in $d = 1$. The present study therefore shows how the Boltzmann equation can be equivalently reobtained from the Euler-scaling limit of the Keldysh field-theoretical description. Equation (114) in $d = 1$ has been also derived in Ref. [55] from the Feynman-Vernon influence functional of the interacting Bose-Hubbard chain. Therein one integrates out the bath degrees of freedom in the system-bath Keldysh action and then takes the limit of strong dissipation (Zeno regime), which eventually renders Eq. (114).

# 6 Dynamics of the lossy Fermi gas

In this Section, we numerically solve the inhomogeneous fermionic Boltzmann equation (114) for a one-dimensional gas. Our aim is to determine the behaviour of the total particle number $N$. In the rescaled Euler coordinates (92), $N$ is obtained as

$$N(\bar{t}) = \int d\bar{x}\, n(\bar{x}, \bar{t}), \quad \text{with} \quad n(\bar{x}, \bar{t}) = \int \frac{dk}{2\pi} n(\bar{x}, k, \bar{t}), \tag{116}$$

where $n(\bar{x}, \bar{t})$ is the spatial density of particles. In the presence of reactions, the dynamics is irreversible. In order to quantify this aspect, we will further calculate the von Neumann thermodynamic entropy $S$. The von Neumann entropy is, indeed, constant in time for unitary-reversible dynamics. In the presence of dissipation it displays, instead, a time dependence $S(\bar{t})$, which quantifies irreversibility. Crucially, in the reaction-limited regime, as discussed in Sec. 5.2, quasi-particles are still well defined and the system in the Euler-scaling limit is described by a time-dependent maximal entropy state of the GGE form. For these maximal entropy states, the von Neumann entropy $S(\bar{t})$ can be calculated on the basis of the knowledge of the Wigner function $n(\bar{x}, k, \bar{t})$ [39,70,94]. For free-fermionic systems, in particular, one has

$$S(\bar{t}) = \int d\bar{x}\, s(\bar{x}, \bar{t}), \tag{117a}$$

with the entropy density

$$s(\bar{x}, \bar{t}) = -\int \frac{dk}{2\pi} \left[ n(\bar{x}, k, \bar{t}) \ln(n(\bar{x}, k, \bar{t})) + (1 - n(\bar{x}, k, \bar{t})) \ln(1 - n(\bar{x}, k, \bar{t})) \right], \tag{117b}$$

which is the von Neumann entropy per unit length of a GGE. In the Euler-scaling limit (92), moreover, the state of the system is locally equivalent at each space-time point $(\bar{x}, \bar{t})$ to a GGE [39,69,70,94,100–102], and therefore $s(\bar{x}, \bar{t})$ represents the density of von Neumann entropy of the reduced density matrix at the space-time point $(\bar{x}, \bar{t})$. The entropy $S$ obtained by integrating $s$ in space thus represents the total entropy of all fluid cells, i.e., the meso-scopic regions around a rescaled space-time point $(\bar{x}, \bar{t})$ which are locally described by a GGE [39,69,70,94,100–107]. In addition, we also note that the expression above for $S$ is, in the context of integrable systems, the Yang-Yang entropy formula for free fermions [108]. This analysis can be therefore considered as an application of the generalised hydrodynamics description of integrable models [100,101,103], to a case with slowly varying potential and weak integrability breaking from dissipation. Furthermore, we will compute the Rényi entropies $S_\alpha(\bar{t})$ of order $\alpha$, which are obtained from the Wigner function according to

$$S_\alpha(\bar{t}) = \int d\bar{x}\, s_\alpha(\bar{x}, \bar{t}), \tag{118a}$$

with

$$s_\alpha(\bar{x}, \bar{t}) = \frac{1}{1 - \alpha} \int d\bar{x} \int \frac{dk}{2\pi} \ln[n^\alpha(\bar{x}, k, \bar{t}) + (1 - n^\alpha(\bar{x}, k, \bar{t}))]. \tag{118b}$$

Here $\alpha$ is an arbitrary positive real number. In the limit $\alpha \to 1$, the expression (118) gives the von Neumann entropy in (117). As in the case of the von Neumann entropy density $s(\bar{x}, \bar{t})$, the Rényi entropy density $s_\alpha(\bar{x}, \bar{t})$ characterises in the Euler-scaling limit the reduced density matrix at the space-time point $(\bar{x}, \bar{t})$. For $\alpha = 2$, namely, $s_2(\bar{x}, \bar{t})$ measures purity of the local GGE state at the space-time point $(\bar{x}, \bar{t})$. The expression (118) has been proved in Refs. [109,110] using the quench action method, whereby it is shown that for non-interacting systems $S_\alpha$ is fixed by the knowledge of $n(\bar{x}, k, \bar{t})$ (this is not the case for interacting systems).

We note that Rényi entropies for open quantum systems have been computed in Refs. [111–113] for free fermionic [111, 112] and bosonic [113] lattice models in the presence of single-body decay (7) and/or creation. In this case the Lindbladian is quadratic and therefore exactly solvable, as outlined in Sec. 4, and the Rényi entropies can then be exactly computed from the knowledge of the two-point fermionic/bosonic correlation function. In the case of binary annihilation of this manuscript, the Lindbladian (88a)-(88b) and (89a)-(89b) is quartic, the dynamics is not exactly solvable and therefore the approach of Refs. [111–113] cannot be pursued. For binary annihilation, the description of the dynamics of the Rényi entropies at the Euler scale necessarily requires considering the large scale description provided by the kinetic equation (110) and (118) or, equivalently, the TGGE ansatz, as we do here.

Namely, we will study the dynamics of the particle number (116) and entropies (117) and (118) in two different scenarios involving a quantum quench of the trapping potential in the presence of binary annihilation dissipation. We consider the instantaneous change, at $t = 0$, of the confining potential, from a "pre-quench" potential $V_0(x)$ to a "post-quench" potential $V(x)$. Both the pre and post-quench potential are taken to be slowly varying in space according to (93). In Subsec. 6.1, we consider a quench from an anharmonic potential $V_0$ to an harmonic confining potential $V$. In Subsec. 6.2, we study the dynamics ensuing from a release of the initial harmonic confinement $V_0$ resulting into the expansion of the gas in free space ($V \equiv 0$). We use henceforth the substitution $J = \frac{\hbar^2}{2m}$. This allows us to connect to the quantum mechanics representation of the kinetic energy, whose eigenvalue is determined by the mass $m$.

## 6.1 Double- to single-well confinement quantum quench

We consider in one spatial dimension a quantum quench of the trapping potential from a pre-quench double-well $V_0(x)$ to a post-quench harmonic well $V(x)$, respectively defined by:

$$V_0(\varepsilon x) = \frac{A}{4}(\varepsilon x)^4 - \frac{m\omega^2}{2}(\varepsilon x)^2 \,, \tag{119a}$$

$$V(\varepsilon x) = \frac{m\omega^2}{2}(\varepsilon x)^2 \,. \tag{119b}$$

Here, we have introduced the adimensional parameter $\varepsilon = \hbar n(0,0)\Gamma/J$. In the reaction-limited regime $\varepsilon \ll 1$, so that both the pre-quench and the post-quench potentials are slowly varying on a large scale set by $\Gamma^{-1}$ according to (92). In this regime, the ground state of the pre-quench potential $V_0$ can be determined with the local density approximation [39]: one treats the quantum gas as consisting of a collection of mesoscopic fluid cells, whose characteristic size is much smaller than the typical length of variation $\ell \sim \Gamma^{-1}$ of the trapping potential $V_0(x)$. Therefore, $V_0(x)$ is assumed to be locally constant and can be reabsorbed into a local chemical potential $\mu - V_0(x)$. Alternatively, one defines a position-dependent dispersion relation $\epsilon_k^{(0)}(x) = \hbar^2 k^2/2m + V_0(x)$ [and, similarly, $\epsilon_k(x) = \hbar^2 k^2/2m + V(x)$ for the post-quench dynamics] for each of the mesoscopic fluid cells at the rescaled space point $\varepsilon x$. We thus determine the initial phase-space distributions $n_0(x,k)$ for the ground state of the Fermi gas in the local density approximation, starting from the Fermi-sea, i.e., we consider the quantum system to be at zero temperature. To this end, we introduce the Fermi-Dirac statistics, whose definition at a generic inverse temperature $\beta$ is given by

$$n_0(\varepsilon x, k, \beta) = \frac{1}{e^{\beta[\epsilon_k^{(0)}(\varepsilon x) - \mu]} + 1} \,, \tag{120}$$

with $\mu$ the chemical potential. In its zero-temperature limit, $\beta \to \infty$, $n_0$ reduces to the Heaviside theta function

$$\lim_{\beta \to \infty} n_0(\varepsilon x, k, \beta) = n_0(\varepsilon x, k) = \begin{cases} 1, & \text{if } \epsilon_k^{(0)}(\varepsilon x) - \mu < 0, \\ 0, & \text{if } \epsilon_k^{(0)}(\varepsilon x) - \mu > 0. \end{cases} \tag{121}$$

Hence, the curve $\zeta = \{(\varepsilon x, k) \in \mathbb{R}^2 \,|\, \mu - \epsilon_k^{(0)}(\varepsilon x) = 0\}$ defines the perimeter of the initial-time occupation function $n_0(\varepsilon x, k)$ in phase space $(\varepsilon x, k)$. Inside $\zeta$, all modes are equally populated and $n_0(x, k) = 1$, i.e., the initial state is locally a Fermi sea. This zero-temperature state (121) therefore has zero entropy (117) since the occupation function $n_0$ takes only the values 0 and 1. In the absence of dissipation, the dynamics ensuing from such zero entropy states has been studied in the framework of zero-temperature generalised hydrodynamics in Ref. [114]. In the presence of binary annihilation dissipation, instead, the dynamics of the Fermi gas with both pre and post-quench harmonic potential has been considered in Ref. [40]. Inserting the definition of $V_0(x)$ given in Eq. (119b), the perimeter $\zeta$ is given by

$$\zeta = \left\{ (\varepsilon x, k) \in \mathbb{R}^2 \,\bigg|\, \mu - \frac{\hbar^2 k^2}{2m} - \frac{A}{4}(\varepsilon x)^4 + \frac{m\omega^2}{2}(\varepsilon x)^2 = 0 \right\}, \tag{122}$$

which corresponds to a bimodal curve (see Figs. 7(a) and 8(a) below). When integrating over momenta, the initial particle density $n_0(x)$ features an initial double-bumped profile.

Once the initial state is determined, we evolve it with the harmonic trap (119b) potential. It is then useful to introduce the dimensionless Euler-scaled space-time coordinates $\tilde{x}, \tilde{t}$ and the rescaled momentum $\tilde{k}$ (cf. also Ref. [40]):

$$\tilde{t} = \varepsilon t \frac{J(2N_0)^{3/2}}{\hbar n(0,0)\ell_{HO}^3}, \qquad \tilde{x} = \frac{\varepsilon}{\sqrt{2N_0}\ell_{HO}} x, \qquad \tilde{k} = \frac{\ell_{HO}}{\sqrt{2N_0}} k, \tag{123a}$$

with $\ell_{HO} = \sqrt{\hbar/m\omega}$ the harmonic oscillator characteristic length. The bimodal initial particle distribution $n_0(\tilde{x}, \tilde{k})$ in the rescaled coordinate is then specified by the condition stemming from Eq. (121)

$$B - \tilde{k}^2 + \tilde{x}^2 - C\tilde{x}^4 > 0, \tag{124}$$

with the parameters $B = \mu/\mu_{HO}$ and $C = A\mu_{HO}/m^2\omega^4$ determining $A$ and $N_0$ (equivalently $A$ and $\mu$). The parameter $\mu_{HO} = N_0\hbar\omega$ is the chemical potential of a gas of $N_0$ particles confined within a harmonic potential. For an harmonic pre-quench potential with frequency $\omega$ therefore $B = 1$, while in our case of (119a) $B \neq 1$. The extremal coordinates of $n_0(\tilde{x}, \tilde{k})$ are given by:

$$\tilde{x}_0 = \sqrt{\frac{1}{2C}\Big[1 + \sqrt{1 + 4BC}\,\Big]}, \qquad \tilde{k}_0 = \sqrt{B}. \tag{125}$$

The Boltzmann equation (114) in the rescaled coordinates is eventually conveniently rewritten as

$$\left[\frac{\partial}{\partial \tilde{t}} + \Omega\Big(\tilde{k}\frac{\partial}{\partial \tilde{x}} - \tilde{x}\frac{\partial}{\partial \tilde{k}}\Big)\right] n(\tilde{x}, \tilde{q}, \tilde{t}) = -\int_{-\infty}^{+\infty} d\tilde{q}\,(\tilde{k}^2 - \tilde{q}^2)\, n(\tilde{x}, \tilde{k}, \tilde{t})\, n(\tilde{x}, \tilde{q}, \tilde{t}), \tag{126}$$

with the adimensional parameter

$$\Omega = \frac{2n(0,0)\ell_{HO}}{(2N_0)^{3/2}} = 2n(0,0)\left(\frac{2J}{8\hbar\omega N_0^3}\right)^{1/2}, \tag{127}$$

expressing the relative strength between coherent motion ($J = \hbar^2/2m$) and confinement ($\omega$).

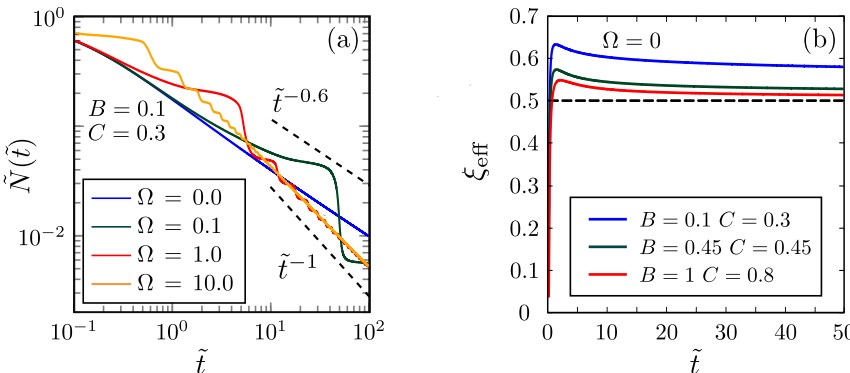

Figure 6: **Double- to single-well confinement quantum quench and decay exponents at $\Omega = 0$.** (a) Decay of the rescaled particle number $\tilde{N}(\tilde{t})$, defined in Eq. (128), in rescaled time $\tilde{t}$ for increasing values of $\Omega$. Increasing $\Omega$ leads to a transition from a power-law decay to an accelerated power-law decay with superimposed oscillations. At long times, all the curves with $\Omega \neq 0$ tend towards a unique curve with decay mean-field-like decay $\tilde{N}(\tilde{t}) \sim \tilde{t}^{-1}$. For the data shown in the figure, we set the parameters in Eq. (124) of the pre-quench anharmonic potential $V_0$ as $B = 0.1$ and $C = 0.3$. (b) Effective decay exponents $\xi_{\text{eff}}$, evaluated by setting $b = 1.1$ in Eq. (130), with parameters $B = 0.1$ and $C = 0.3$, $B = 0.45$ and $C = 0.45$, $B = 1$ and $C = 0.8$, respectively from top to bottom, and $\Omega = 0$. As the ratio $B/C$ decreases, entailing that $\tilde{k}_0$ in Eq. (125) decreases and that the initial distribution is increasingly deformed with respect to the circular-harmonic shape, the long-time effective decay exponent $\xi_{\text{eff}} > 1/2$ shows a power-law decay accelerated with respect to the homogeneous prediction $\tilde{N}(\tilde{t}) \sim \tilde{t}^{-1/2}$.

As written above, we are interested in the dynamics as a function of time of the total particle number (116) and the von Neumann (117) and Rényi entropies (118). In the adimensional Euler-scaled coordinates $\tilde{x}$ and $\tilde{t}$ (123a), we accordingly define the rescaled particle number $\tilde{N}(\tilde{t})$ and entropies $\tilde{S}_\alpha$

$$\tilde{N}(\tilde{t}) = \frac{N(\tilde{t})}{N_0} = \frac{\int d\tilde{x}\, d\tilde{k}\, n(\tilde{x}, \tilde{k}, \tilde{t})}{\int d\tilde{x}\, d\tilde{k}\, n(\tilde{x}, \tilde{k}, \tilde{0})}, \tag{128a}$$

$$\tilde{S}_\alpha(\tilde{t}) = \frac{\pi \varepsilon}{N_0} S_\alpha(\tilde{t}) = \frac{1}{1-\alpha} \int d\tilde{x}\, d\tilde{k}\, \log\left[ n(\tilde{x}, \tilde{k}, \tilde{t})^\alpha + (1 - n(\tilde{x}, \tilde{k}, \tilde{t}))^\alpha \right]. \tag{128b}$$

When reactions are not present $I_{\text{coll}}[n(\tilde{x}, \tilde{k}, \tilde{t})] = 0$, the kinetic equation can be solved analytically using the method of characteristics. The characteristic equations

$$\begin{cases} \dot{\tilde{x}} = \Omega \tilde{k}, \\ \dot{\tilde{k}} = -\Omega \tilde{x}, \end{cases} \tag{129}$$

define harmonic circular trajectories in the rescaled phase space, with rescaled period $\tilde{T} = 2\pi/\Omega$. When two-particle losses are included, the solution is determined by both phase-space rotations and by the non-vanishing collision integral on the right hand side of Eq. (126). The numerical solution to Eq. (126) allows us to identify two regimes in the decay of $\tilde{N}(\tilde{t})$ as a function of $\tilde{t}$ depending on the chosen value of $\Omega$. We show this in Fig. 6(a)-(b). In the limiting regime where $\Omega \ll 1$, the evolution along characteristic trajectories of the Wigner function is suppressed. This corresponds to the decay for $\Omega = 0$ in Fig. 6(a). Therein, the dominant term in determining the dynamics of the density distribution is the collision integral. Consequently, we expect the decay to be slower than in the bosonic mean-field case, as an effect of the $(\tilde{k} - \tilde{q})^2$

factor, which further limits reactions to occur when particles have similar momenta. In this limit, we, indeed, find power-law decay, as quantified by the effective exponent $\xi_{\text{eff}}$ [6]:

$$\xi_{\text{eff}} = -\frac{\log\left[\tilde{N}(b\tilde{t})/\tilde{N}(\tilde{t})\right]}{\log(b)} . \tag{130}$$

If $\tilde{N}(\tilde{t})$ asymptotically approaches a power law, namely $\tilde{N}(\tilde{t}) \sim \tilde{t}^{-\xi}$ at long times, then $\lim_{t\to\infty} \xi_{\text{eff}} = \xi$, for any value of $b$. Here, $b$ is a scaling parameter. In the numerical evaluation of Eq. (130), we use henceforth $b = 1.1$. In Fig. 6(b), we, indeed, see that for $\Omega = 0$, the decay effective exponents converges at long time. The asymptotic value, interestingly depends on the parameters $B$ and $C$ characterising the anharmonicity of the initial state. In the case when $B$ and $C$ are taken such that the extremal coordinates Eq. (125) satisfy $\tilde{x}_0 \sim \tilde{k}_0$, the initial bimodal distribution is approximately circular (cf. also Fig. 7(a) below) and one obtains $\tilde{N}(\tilde{t}) \sim \tilde{t}^{-1/2}$. This case is achieved in Fig. 6(b) for $B = 1$ and $C = 0.8$. This observation matches the result of Ref. [40, 56]. In Ref. [40], in particular, it has been analytically shown that for an initial harmonic $V_0(x)$ potential and $\Omega = 0$, the decay exponent is identical to the homogeneous case (115). This can be understood since when both $V_0$ and $V$ are harmonic, taking the limit $\Omega \to 0$, amounts to considering $N_0 \gg 1$ according to Eq. (127). In the limit $N_0 \gg 1$, the initial density profile around its maximum at $\tilde{x} = 0$, where most of the reactions take place, becomes approximately flat and homogeneous. The homogeneous decay exponent (115) is therefore recovered. In the cases $B = 0.45, C = 0.45$ and $B = 1, C = 0.8$, however, the anharmonicity of the initial state is more pronounced (cf also Fig. 8(a) below). Here, the initial density distribution does not become flat in the limit $\Omega \to 0$ and therefore the effect of the initial density inhomogeneity cannot be neglected. This eventually causes the decay $\tilde{N}(\tilde{t}) \sim \tilde{t}^{-\xi}$ in Fig. 6(b) with $1/2 < \xi_{\text{eff}} < 1$. We therefore conclude that anharmonicity of the initial state causes a faster decay for small $\Omega$ than in the case where also the pre-quench potential $V_0$ is harmonic.

As $\Omega$ increases, we can see from Fig. 6(a) that oscillations are superimposed to the power-law decay. These oscillations are understood because of phase-space rotations (c.f. Figs. 7(b) and 8(b) below). At short times the gas mostly depletes at the space points corresponding to the two minima of the double well (Fig. 7(b)), where initially most particles are located. Simultaneously, long-lived modes with non vanishing $\pm k$ are formed on the right (left) of the trap $\tilde{x}$ ($\tilde{x} < 0$). These modes travel towards the edges of the trap and bounce back. In this way, the density of particles gets peaked around the centre $\tilde{x} = 0$. When the two counter-propagating modes meet at the centre of the trap they lead to an acceleration of the decay. This happens at time which is approximately $\tilde{t} \sim \pi/\Omega$. The resulting breathing motion of the quantum gas renders particle losses periodically accelerated at period $\tilde{T} \sim \pi/\Omega$. At long times, for any $\Omega \neq 0$, the curves $\tilde{N}(\tilde{t})$ collapse towards a unique power-law with mean-field-like behaviour $\tilde{N}(\tilde{t}) \sim \tilde{t}^{-1}$. Remarkably, both for both considered cases $B = 1$, $C = 0.8$, and $B = 0.1$, $C = 0.3$, as shown in Fig. 6. The asymptotic mean-field-like decay therefore does not depend on how much the pre-quench potential $V_0(x)$ deviates from an harmonic shape. A similar transition to an accelerated oscillatory decay has been, indeed, also observed in [40] for an initial harmonic pre-quench potential $V_0$. In that case, the transition to the accelerated decay takes place at times $\tilde{t} \sim \pi/(2\Omega)$. For $\Omega$ large, therefore, one has asymptotic mean-field decay for both harmonic and anharmonic pre-quench potentials.

In Fig. 7, we report the dynamics of the rescaled entropies (128) as a function of the rescaled time $\tilde{t}$ for different values of $\alpha = 1, 2, 3$. In Fig. 7(c), specifically, we consider the case $\Omega = 0$, where the contour in Fig. 7(a) does not undergo rotations in phase space. We take the parameters $B = 1$ and $C = 0.8$ so that the initial contour in Fig. 7(a) is approximately circular. This corresponds to a prequench potential $V_0(x)$ weakly deviating from the harmonic shape. We observe that all the rescaled entropies $\tilde{S}_\alpha(\tilde{t})$ have similar dynamics as a function of

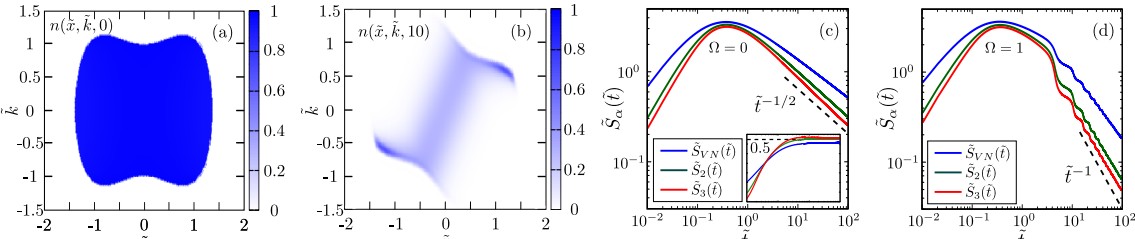

Figure 7: **Weak anharmonicity — double-to single-well confinement quench and entropy decay.** (a)-(b) Wigner distribution $n(\tilde{x}, \tilde{k}, \tilde{t})$ in the rescaled $(\tilde{x}, \tilde{k})$ phase space, at selected rescaled times $\tilde{t} = 0, 10$, respectively. In both panels, we set $\Omega = 0.1$, in Eqs. (126) and (127). In the initial state distribution we set $B = 1$, $C = 0.8$. For this choice of parameters, the extremal coordinates (125) of the initial distribution $\tilde{x}_0 \sim \tilde{k}_0$ are approximately equal and the distribution is approximately circular-harmonic in phase space. (c) Rescaled Rényi entropies $\tilde{S}_\alpha(\tilde{t})$ for $\alpha = 1, 2, 3$, defined in Eq. (128), as function of the rescaled time $\tilde{t}$ for $\Omega = 0$. In the inset: effective exponents of the curves in panel (c). All the Rényi entropies asymptotically decay as $\tilde{S}_\alpha(\tilde{t}) \sim \tilde{t}^{-1/2}$. (d) Rescaled Rényi entropies $\tilde{S}_\alpha(\tilde{t})$ as a function of $\tilde{t}$ for $\Omega = 1$. In this case an accelerated oscillatory decay is observed. At long times, all the entropies display decay $\tilde{S}_\alpha(\tilde{t}) \sim \tilde{t}^{-1}$ with mean field exponent.

$\tilde{t}$ independently of the Rényi index $\alpha$. The entropies at $\tilde{t} = 0$ are zero since the initial state (121) is a ground state. As time progresses, dissipation scrambles the occupation in phase space rendering $0 < n(\tilde{x}, \tilde{k}, \tilde{t}) < 1$ within the Fermi contour. This corresponds to an increased mixedness of the local GGE state at the space-time point $(\tilde{x}, \tilde{t})$, as further witnessed by the Rényi entropy $\tilde{S}_2$, which quantifies the local purity of the GGE state at the space-time point $(\tilde{x}, \tilde{t})$. At longer times, dissipation clearly drives the system towards the vacuum (since particles can only be lost) and therefore the state gets purified and $\tilde{S}_\alpha(\tilde{t})$ decreases after reaching a maximum value. From the numerical calculation of the associated effective exponent (130), see inset of Fig. 7(c), we quantify the asymptotic decay in time as a power law $\tilde{S}(\tilde{t}) \sim \tilde{t}^{-1/2}$. The decay exponent is therefore the same as that found for the density in Fig. 6 for the same choice of parameters $B = 1$ and $C = 0.8$. For $\Omega > 0$, as in Fig. 7(d), the breathing motion of the fermionic gas induces oscillations, as in the case of the density decay. Moreover, the entropies, for all values of $\alpha$, decay at long times with a mean-field exponent $\tilde{S}_\alpha(\tilde{t}) \sim \tilde{t}^{-1}$.

We note that the fact that the entropies, in particular the von Neumann entropy $S(\tilde{t}) \equiv S_1(\tilde{t})$ (117), are not monotonic as a function of time, does not contradict the second law of thermodynamics. In the context of open quantum systems weakly coupled to a reservoir at thermal equilibrium at temperature $T$, entropy production $\sigma$ is given, as shown, e.g., in Refs. [115–119], by

$$\sigma(t) = \frac{dS(t)}{dt} + \mathcal{J}, \quad \text{with} \quad \mathcal{J} = -\frac{1}{T}\frac{d\langle H \rangle}{dt}. \tag{131}$$

The entropy production $\sigma \geq 0$ is proven [115–117] to be nonnegative, which is the statement of the second law of thermodynamics. The entropy production thus represents the total amount of entropy produced per unit time in the system and environment due to the irreversibility of the dynamics. The two terms on the right hand side of (131), individually, are not bound to be nonnegative. The first term is the time derivative of the system von Neumann entropy. This is precisely the contribution we have computed. The second term $\mathcal{J}$ is the entropy exchanged between the system and the environment due to exchange of heat. Here, this quantity is positive, so entropy flows from the system to the environment, since the energy in the system is monotonically decreasing. The system's energy decreases as a consequence of

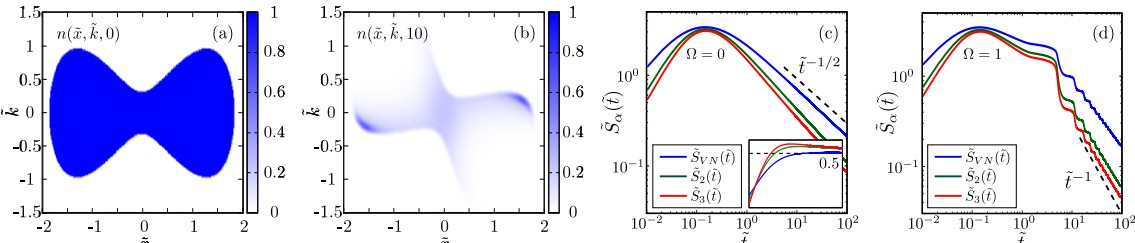

Figure 8: **Strong anharmonicity — double- to single-well confinement quench and entropy decay.** (a)-(b) Particle distribution $n(\tilde{x}, \tilde{k}, \tilde{t})$ in the rescaled $(\tilde{x}, \tilde{k})$ phase space, at selected rescaled times $\tilde{t} = 0, 10$, respectively. In both panels, we set $\Omega = 0.1$. In the initial state distribution, we set $B = 0.1$, $C = 0.3$. The initial distribution therefore displays a pronounced bimodal shape significantly deviating from the harmonic-circular shape. (c) Rescaled Rényi entropies $\tilde{S}_\alpha(\tilde{t})$ as a function of the rescaled time $\tilde{t}$ for Rényi index values $\alpha = 1, 2, 3$ (from top to bottom) and $\Omega = 0$. All the Rényi entropies decay asymptotically as $\tilde{S}_\alpha(\tilde{t}) \sim \tilde{t}^{-1/2}$ as in the case of Fig. 7. (d) Rescaled Rényi entropies $\tilde{S}_\alpha(\tilde{t})$ as a function of $\tilde{t}$ for $\Omega = 1$. All the entropies decay as $\tilde{S}_\alpha(\tilde{t}) \sim \tilde{t}^{-1}$ at long times with superimposed oscillations.

the loss in time of particles. Since particles are only lost, the environment behaves as a zero temperature bath, $T \to 0$ in Eq. (131), and therefore $\mathcal{J} \to +\infty$. This divergence of $\mathcal{J}$, in the presence case of dissipative losses, makes $\sigma \geq 0$ even if the change of system entropy $dS/dt$ can be negative (due to purification of the state as explained above).

In Fig. 8, we discuss the dynamics of the rescaled entropies $\tilde{S}_\alpha(\tilde{t})$ as a function of $\tilde{t}$ for a different choice of the parameters $B = 0.1$ and $C = 0.3$ characterising the initial anharmonic potential. For this choice of parameters the initial distribution in Fig. 8(a), displays a pronounced bimodal profile. In Fig. 8(c), for $\Omega = 0$, we observe that all the entropies decay asymptotically $\tilde{S}_\alpha(\tilde{t}) \sim \tilde{t}^{-1/2}$ (the effective exponents are reported in the inset). For the entropy, therefore, the anharmonicity of the pre-quench potential does not modify the asymptotic decay exponent, in contrast to from the previously discussed case of the density. In Fig. 8(d), we consider the case $\Omega = 1$, where rotations in phase space of the Fermi contour (cf. Fig. 8(b)) are present. In this case, we again find, similarly to Fig. 7, that the breathing motion of the gas lead to an eventual accelerated decay $\tilde{S}(\tilde{t}) \sim \tilde{t}^{-1}$.

## 6.2 Deconfinement from the harmonic trap

As a second example, we study the dynamics of the Fermi gas initially confined by a pre-quench harmonic potential

$$V_0(\varepsilon x) = \frac{m\omega^2}{2}(\varepsilon x)^2, \tag{132}$$

with $\varepsilon$ given by the expression shown after Eq. (119b). At time $t = 0$, the confining potential is suddenly switched off, so that the initial bump density profile is allowed to freely expand in space (post-quench potential $V(x) \equiv 0$). As no external potential affects the post-quench dynamics, the Boltzmann equation (114) takes the simplified form (in the dimensionful Euler-scaled variables $\bar{x}, \bar{t}$ (92))

$$\left[ \frac{\partial}{\partial \bar{t}} + \frac{\hbar k}{m} \frac{\partial}{\partial \bar{x}} \right] n(\bar{x}, k, \bar{t}) = -\int \frac{dq}{2\pi} (k-q)^2 n(\bar{x}, k, \bar{t}) n(\bar{x}, q, \bar{t}). \tag{133}$$

However, the initial confining trap $V_0$ determines the shape and the size of the initial distribution $n_0(\varepsilon x, k, \beta)$. In the local density approximation, the ground state is obtained by taking the

zero-temperature ($\beta \to \infty$) limit of the Fermi-Dirac distribution, with $\epsilon_k^{(0)}(\varepsilon x)$ determined by the harmonic potential. The perimeter $\zeta$ of the initial bump distribution is therefore an ellipse in the $(\varepsilon x, k)$ plane:

$$\zeta = \left\{ (\varepsilon x, k) \in \mathbb{R}^2 \,\middle|\, \mu - \frac{\hbar^2 k^2}{2m} - \frac{m\omega^2}{2}(\varepsilon x)^2 = 0 \right\}. \tag{134}$$

Then, $n_0(\varepsilon x, k) = 1$ inside $\zeta$, i.e., for $\epsilon_k^{(0)}(\varepsilon x) < \mu$, while it is vanishing otherwise. Integrating over the surface internal to $\zeta$ (134), the chemical potential can be evaluated to be $\mu = N_0 \hbar \omega$. At this point, we defined dimensionless Euler-scaled variables $\tilde{x}$ and $\tilde{t}$ as in Eq. (123a). The momentum is similarly rescaled as in (123a). Thus, the ellipse contour in (134) turns into a circle

$$\tilde{x}^2 + \tilde{k}^2 < 1, \tag{135}$$

as shown in Fig. 9(a). Besides, Eq. (133) takes the rescaled form:

$$\left[ \frac{\partial}{\partial \tilde{t}} + \Omega \tilde{k} \frac{\partial}{\partial \tilde{x}} \right] n(\tilde{x}, \tilde{k}, \tilde{t}) = - \int_{-\infty}^{+\infty} d\tilde{q} \, (\tilde{k}^2 - \tilde{q}^2) n(\tilde{x}, \tilde{k}, \tilde{t}) n(\tilde{x}, \tilde{q}, \tilde{t}), \tag{136}$$

with $\Omega$ given in Eq. (127). The rescaled particle number $\tilde{N}$ and entropies $\tilde{S}$ are also as in Eq. (128).

Once more, the (136) can be solved numerically via the method of characteristics. The set of equations defining the characteristic trajectories is given by

$$\begin{cases} \dot{\tilde{x}} = \Omega \tilde{k}, \\ \dot{\tilde{k}} = 0. \end{cases} \tag{137}$$

Momenta remain constant throughout the full evolution, meaning that momentum modes do not get mixed, as it happened in the case of harmonic confinement. Conversely, position coordinates evolve via the simple relation $d\tilde{x}/d\tilde{t} = \Omega \tilde{k} = \Omega \tilde{k}_0$ identifying straight horizontal lines in phase space, i.e., the trajectories are parallel to the $\tilde{x}$ axis. Hence, in absence of particle losses the initial circle is stretched into an elongated disc, as shown in Fig. 9(b): particles located on the $\tilde{x}$ axis stay still, while extremal modes $\tilde{k}_0 = \pm 1$ slide with velocity $\pm \Omega$. This corresponds to the initial distribution of particles spreading in two opposite directions. The parameter $\Omega$ then plays the role of an "escape" velocity. By varying its value one can still identify two well-separated decay behaviours for the density, as discussed in Ref. [56].

We briefly review here the results of Ref. [56] for the decay as a function of time of the density and how it depends on $\Omega$. In this manuscript, we then additionally focus on the dynamics as a function of time of the von Neumann and Rényi entropies. In particular, we study the entropy dynamics in the trap-release quench for different values of $\Omega$.

In the limiting case $\Omega \to 0$, the evolution along the characteristic curves is once more suppressed, and one obtains the same decay as in the homogeneous setting (115), i.e., the algebraic decay $\tilde{N}(\tilde{t}) \sim \tilde{t}^{-1/2}$. Increasing the value of the escape velocity $\Omega$, the system approaches an unexpectedly slow decay. As particles propagate in free space, the local spatial density decreases. Consequently, the Wigner function $n(\tilde{x}, \tilde{k}, \tilde{t})$ for each value of $\tilde{x}$ becomes supported on a narrow interval of $\vec{k}$ values, as one can see from Fig. 9(b). Reactions are constrained to take place between particles with the same momentum. This kind of reactions is further suppressed by the fermionic statistics, which manifest in the blocking factor $(\tilde{k} - \tilde{q})^2$ in the collision integral. As a result, for every finite value of $\Omega > 0$, at short times one has an approximate algebraic decay $\tilde{N}(\tilde{t}) \sim \tilde{t}^{-\xi}$. The exponent $\xi$ gets smaller and smaller as $\Omega$ increases. At longer times, a slower non-algebraic decay. Such non-algebraic decay is a property specific to the Fermi gas due to the blocking factor in the collision integral of Eq. (136).

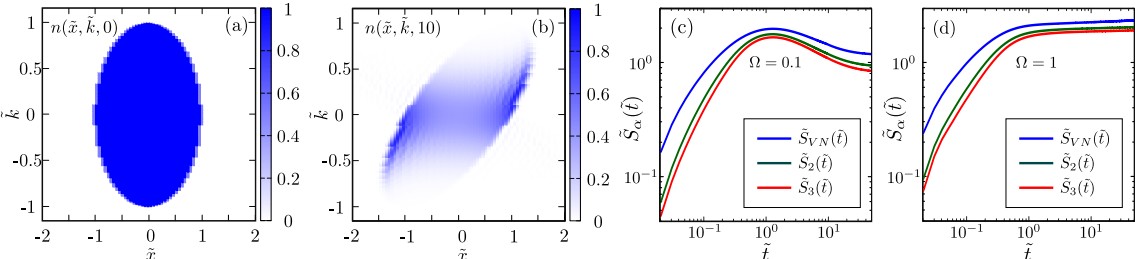

Figure 9: **Entropy dynamics in a trap-release quench of the Fermi gas with binary annihilation.** (a)-(b) Particle distribution $n(\tilde{x}, \tilde{k}, \tilde{t})$ in the rescaled $(\tilde{x}, \tilde{k})$ phase space, at selected times $\tilde{t} = 0, 10$, respectively. We set $\Omega = 0.1$. (c) Rescaled Rényi entropies as a function of $\tilde{t}$ for Rényi indices $\alpha = 1, 2, 3$ (from top to bottom) and $\Omega = 0.1$. All the entropies first increase in time and reach a maximum value, after which a slow non-algebraic decay is established. (d) Rescaled entropies as a function of rescaled time for $\Omega = 1$. Here, the entropies first increase in time as a consequence of reactions, and then saturate to an approximately constant value as a consequence of ballistic-reversible spreading in free space.

In Fig. 9(c)-(d), we report the dynamics of the rescaled entropies $\tilde{S}_\alpha(\tilde{t})$ as a function of the rescaled time $\tilde{t}$. In Fig. 9(c), instead, we take $\Omega = 0.1$, which allows to identify two regimes also for the entropy dynamics, similarly to the aforementioned discussed case of the density. First the entropy increases in time and reaches a maximum value. This corresponds to the regime where particles are still concentrated around the centre of the trap and reactions are frequent. The density decays in time as a power-law as a consequence of the reactions and the entropy, instead, increases due to the increased mixedness of the local state describing each fluid cell at the rescaled point $\tilde{x}, \tilde{t}$. As the gas expands in free space, reactions become scarcer and scarcer. Consequently, the slower non-algebraic decay of the density translates into a similar slower non-algebraic decay for the entropy. The entropy slowly decays towards zero, which signals the slow, non-algebraic approach of the system towards the vacuum. Also in this case, as commented after Eq. (131), the entropy decrease is not in contradiction with the second law of thermodynamics since the entropy flux $\mathcal{J}$ due to heat exchange between the system and the environment is divergent (zero-temperature bath). We observe that for $\Omega = 0$, the behaviour of the rescaled entropies is similar to the one discussed in Fig. 7(c). In particular, the slow, non-algebraic, asymptotic decay is not present in this case. On the contrary, the entropies decay as $\tilde{S}_\alpha(\tilde{t}) \sim \tilde{t}^{-1/2}$ asymptotically in time. In Fig. 9(d), we, instead, consider the case of a larger $\Omega$ value ($\Omega = 1$). The entropies increase monotonically in time and eventually saturate to a constant. The saturation is caused by the fact that for large $\Omega$ the spreading of particles in free space is so rapid that reactions are almost completely suppressed due to the fermionic blocking factor $(k - q)^2$ recalled above. The dynamics is therefore governed by the ballistic propagation of the gas in free space. This is precisely the left hand side of Eq. (136), while the collision integral can be approximately neglected (the system can be considered as being approximately isolated). The ensuing Euler equation is time reversible [69, 70] and therefore it does not lead to production of entropy (both the von Neumann (117) and Rényi (118) entropy are conserved under Euler evolution). The behaviour of the entropies in Fig. 9(d) is therefore intrinsically related to the the fact that in the present quantum RD models transport is ballistic. In classical RD systems, instead, inhomogeneities in the initial state are smoothed out by diffusion, which is irreversible, resulting in a further growth of entropy.

# 7 Conclusions

In this manuscript, we elaborated and further developed in details the results of the work [56], considering many-body fermionic and bosonic gases subject to dissipative loss processes. The ensuing quantum reaction-diffusion dynamics, where quantum coherent motion replaces classical diffusion, was formulated in terms of the Markovian quantum master equation in the Lindblad form. We discussed these models in continuum space, with the Hamiltonian term (12) describing Hamiltonian-coherent quantum transport, and the jump operators (13)-(15) modelling irreversible reactions. Exploiting the Keldysh path-integral representation of the quantum master equation, summarised in Sec. 3, we obtained a field theory formulation of the dynamics. Importantly, the partition function $Z$ in Eq. (22) contains both a bulk contribution with the Keldysh action $S$ and additional boundary terms. In the latter, the fields are computed only at the initial time $t_0$ ($t_0 = 0$ throughout the manuscript). While the bulk Keldysh action $S$ describes the stationary state of the dynamics, boundary terms account for the dynamical approach to the steady state.

We explicitly showed this aspect in Sec. 4, where we benchmarked the inclusion of the boundary terms in the exactly solvable case of one-body decay $A \to \emptyset$. We studied both pure initial states, such as the initial state in Eq. (49) where only the mode $k = 0$ is initially occupied, and thermal initial conditions of (generalised) Gibbs form (67). In both cases, we showed that the inclusion of the boundary terms changes the Keldysh Green's function $G_{0,S}^K$, which depends on the initial occupation function. In particular, the Keldysh Green's function, cf. Eq. (63) associated to (49) and Eq. (78) associated to (67), is not time-translation invariant in the presence of boundary terms. From this result, we retrieved exponential decay (65) of the particle density towards the stationary state devoid of particles. In Subsec. 4.3, we also discussed the effect of the initial-boundary terms on the normalisation of the partition function $Z$. We did this by means of a perturbative expansion of the initial-boundary terms with respect to the bulk Gaussian weight $S_0$ (56) determined by coherent hopping and decay. This expansion further confirms that $G_{0,S}^K$ is not translationally invariant. Furthermore, we found that the normalisation of the partition function $Z$ is $Z = \mathcal{N}$, where $\mathcal{N}$ is the normalisation of the initial state (see Eqs. (49) and (67)). This normalisation is therefore different from that of the bulk Keldysh action (25) $Z_K = 1$, where no boundary terms are present [64–67].

In Sec. 5, we then moved to considering the interacting, not exactly solvable, case of binary annihilation $A + A \to \emptyset$. We consider the dynamics in the so-called reaction-limited regime of weak dissipation, i.e., small $\Gamma$. Since dissipation determines the quartic interaction vertices with coupling $\Gamma$ in (88a) (bosons) and (89a) (fermions), the reaction-limited regime can be tackled by means of perturbative expansions. First, in Subsec. 5.1, we showed that the perturbative expansion of the interaction vertices does not alter the normalisation of the Keldysh partition function $Z_K = 1$ (and therefore also of the partition function $Z = \mathcal{N}$). Subsequently, in Subsec. 5.2, we described how a kinetic description of the quantum RD dynamics in the form of a Boltzmann equation can be derived in the reaction-limited regime. The derivation is based on the Euler-scaling limit of hydrodynamics (92) and (93). In the Euler scaling limit, the Moyal derivative expansion of the kinetic equation can be truncated at first order in space-time derivatives and the kinetic equation takes the form a Boltzmann equation (101). In this equation, the collision integral is given by dissipation, and we compute it in the Euler-scaling limit from the tadpole diagrams in Fig. 4 (bosons) and 5 (fermions). The main result of the analysis is eventually given by the kinetic equation (110) (bosons) and (114) (fermions). For bosons in homogeneous setups, one finds algebraic decay with mean-field exponent (112) in all spatial dimensions. For fermions, instead, algebraic decay (115) with exponent different from the mean-field one is observed in all dimensions, as already pointed out in Ref. [56]. In Sec. 6, we eventually specialise the analysis to fermions in one spatial dimension in the pres-

ence of a trapping potential. We discuss the experimentally relevant case of quenches of the trapping potential. We find that for an initial anharmonic potential, in Fig. 6, the decay of the particle density is accelerated compared to both the case of translationally invariant systems (115) and harmonic confinement. We then characterise the irreversibility of the dynamics due to dissipation by computing the time-dependence of the von Neumann (117) and Rényi entropies (118). We find that for quantum quenches from an anharmonic to harmonic potential, in Figs. 7 and 8, the entropy decays asymptotically in time according to a power law with the same exponent as the density. Specifically, for small coherent hopping/strong confinement, the entropy decays algebraically in time with non mean-field exponent, see Figs. 7(c) and 8(c). Here, the corresponding decay exponent is not changed upon tuning the anharmonicity of the initial potential. For strong coherent hopping/weak confinement, instead, the entropies decays algebraically with mean-field exponent, see Figs. 7(d) and 8(d). In all cases, the decay of the entropy comes from the asymptotic cooling of the gas due to heat exchange with the surrounding zero-temperature bath (since particles can only be lost from the system). For the different trap-release quench, in Fig. 9, we, instead, observe that entropy monotonically increases in time and eventually saturates. This saturation is explained in terms of the underlying quantum ballistic motion, which is described by the Euler equation. The latter does not contain viscosity terms and it is therefore reversible, thus preventing any entropy production.

As a future perspective, it would be interesting to study the case where also Hamiltonian interactions are present. For example, either by introducing contact interactions in the Bose gas [120], or by considering spinful Fermi models such as the mass-imbalanced Fermi-Hubbard model [121]. For weak interactions the treatment in terms of the kinetic-Boltzmann equation still applies. The presence of Hamiltonian interactions, in general, leads to a breaking of the integrability of the Hamiltonian and results in hydrodynamic diffusion of the remaining conserved charges, as shown in Ref. [122]. It is then natural to wonder, for example by considering the particle density, how quantum diffusive transport can affect the asymptotic decay law and the associated exponents compared to the case of ballistic transport discussed here. At the same time, it is also interesting to look at the effect of Hamiltonian integrability breaking on the entropy dynamics. Moving away from the regime of weak dissipation (reaction-limited), it is of fundamental importance to understand the quantum analogue of the diffusion-limited regime, for which currently there are no analytical predictions. This regime cannot be treated via the kinetic-Boltzmann equation and it requires a renormalisation group analysis. As shown in Refs. [8–10, 22, 23] for classical RD, specifically, one needs to perturbatively expand both the initial-boundary terms and the interaction vertices to identify the renormalisation group flow of the coupling constants. In the present quantum case, we expect that quantum diffusion-limited binary annihilation $A+A \to \emptyset$ can be similarly tackled by including the initial-boundary terms as explained in Sec. 4. In this sense, it is also particularly interesting to compare the asymptotic behaviour of fermions and bosons. While in the classical case occupancy restrictions do not affect the asymptotic diffusion-limited decay (since the particle density is already low), in the quantum case it is not immediate to understand whether bosons and fermions could yield the same decay law. We leave the analysis of these important points for future studies.

## Acknowledgments

We thank F. Carollo for fruitful discussions on the concept of entropy production in open quantum systems.

**Funding information** F.G. thanks Universität Tübingen for hospitality, and acknowledges support from Università di Trento and Collegio Bernardo Clesio. G.P. acknowledges support from the Alexander von Humboldt Foundation through a Humboldt research fellowship for postdoctoral researchers. We also acknowledge funding from the Deutsche Forschungsgemeinschaft (DFG, German Research Foundation) through the Research Unit FOR 5522, Grant No. 499180199.

# A Classical reaction-diffusion field theory for binary annihilation

In this Appendix, we briefly discuss the field theory formulation of classical binary annihilation $A + A \to \emptyset$. This is achieved via the Doi-Peliti path integral representation of the classical master equation [8–10, 22, 23]. For the purpose of this work, we just report here the Doi-Peliti action for classical $A + A \to \emptyset$ for the sake of comparison with the quantum Keldysh description in Eqs. (88a)-(89b). As a matter of fact, a full description of the renormalisation programme within classical RD models is beyond the scope of this work, and we refer to [8–10, 22, 23] for a detailed discussion. For classical binary annihilation $A + A \to \emptyset$ the Doi-Peliti action is

$$S[\varphi, \tilde{\varphi}] = \int_0^\infty dt \int d^d x \, \tilde{\varphi} \, (\partial_t - D \nabla^2) \varphi + \left[ 2\Gamma \tilde{\varphi} \varphi^2 + \Gamma \tilde{\varphi}^2 \varphi^2 - n_0 \tilde{\varphi} \, \delta(t) \right]. \qquad \text{(A.1)}$$

This action describes a system where multiple occupancy of the same lattice site is allowed and therefore is akin to the bosonic quantum RD formulation (88a). The field fields $\varphi$ and $\tilde{\varphi}$ are, indeed, complex valued fields associated to the eigenvalues of the destruction and creation operators, respectively, introduced in the coherent-state path integral description of the classical master equation. In particular, $\varphi$ is related to the mean density $n = \langle \varphi \rangle$, while $\tilde{\varphi}$ is related to complex conjugate field $\bar{\varphi}$ through the so-called Doi shift $\tilde{\varphi} = 1 + \bar{\varphi}$. In the classical case, therefore, the mean density is linear in the field, while in the quantum case is quadratic as discussed in Eqs. (32) and (37). This is also consistent with the different engineering dimensions of $\varphi \sim \ell^{-d}$ and $\tilde{\varphi} \sim \ell^0$ in the continuum limit compared to the quantum symmetric case of (10). The quadratic part of the action (A.1) reflects classical stochastic diffusive motion with diffusion constant $D$. In the quantum case in Eq. (88b), an imaginary factor is present in front of the time derivative reflecting the different ballistic transport present in the quantum case, with coherent hopping amplitude $J$. In both the cases, the quadratic part of the action leads to a fully off-diagonal propagator. For the classical action (A.1), the Green's function associated to the quadratic part are

$$\langle \varphi(\vec{x}, t) \tilde{\varphi}(\vec{x}', t') \rangle = \Theta(t - t') \left[ \frac{1}{4\pi D(t - t')} \right]^{d/2} \exp\left[ -\frac{(x - x')^2}{4D(t - t')} \right], \qquad \text{(A.2)}$$

$$\langle \tilde{\varphi}(\vec{x}, t) \varphi(\vec{x}', t') \rangle = \Theta(t' - t) \left[ \frac{1}{4\pi D(t' - t)} \right]^{d/2} \exp\left[ -\frac{(x - x')^2}{4D(t' - t)} \right], \qquad \text{(A.3)}$$

which are the classical analogue of $G^R$ and $G^A$ in Eq. (57) ($\Gamma_d = 0$), respectively. For binary annihilation $\langle \varphi(\vec{x}, t) \varphi(\vec{x}', t') \rangle = \langle \tilde{\varphi}(\vec{x}, t) \tilde{\varphi}(\vec{x}', t') \rangle \equiv 0$ and therefore there is no classical analogue of the Keldysh Green's function (36c). The boundary term proportional to $\delta(t)$ reflects the Poissonian initial condition which is typically taken in classical RD. In this initial condition, each lattice site is occupied by a number of particles distributed according to a Poissonian with parameter $n_0$. The latter is the same for each lattice site and hence the initial state is homogeneous. The coupling to the initial density $n_0$ is linear in the fields $\tilde{\varphi}$ in a similar way to Eq. (50). In the quantum case, since the coefficients of the series in (48) are, however, amplitudes (not probabilities) the parameter associated to the state is $\sqrt{n_0}$.

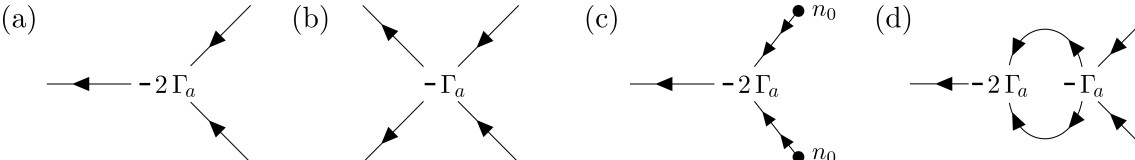

Figure 10: **Interaction vertices for classical $A + A \to \emptyset$.** Here in the diagrams, time flows from right to left, as indicated by arrows on the fields. Both vertices (a) and (b) feature two incoming $\varphi$ fields, that are used to connect to earlier sources via fields $\tilde{\varphi}$. Vertex (a) has only one "surviving" outgoing $\tilde{\varphi}$ field, while (b) has two $\tilde{\varphi}$ fields. Panel (c) represents the Y-shaped cubic diagram where initial-time sources $n_0$ have been expanded to second-order in perturbation theory in $n_0$, and connected to vertex (a) via two bare propagators (A.2) and (A.3). (c) thus provides the starting point for the tree-level Dyson equation for the tree-level density $n_{\text{tree}(t)}$. The latter is obtained by nesting vertex (a) recursively. By combining (a) and (b) together, one defines the fundamental diagram (d) providing the one-loop correction to $n_{\text{tree}}$. The loop series is thus given by repeated iteration of (b).

Considering the interacting – non-quadratic – part of the action, we give in Fig. 10 a pictorial representation of the interaction vertices $2\Gamma\tilde{\varphi}\varphi^2$ and $\Gamma\tilde{\varphi}^2\varphi^2$. The interaction vertices in Eq. (A.1) are both cubic (Fig. 10(a)) and quartic (Fig. 10(b)) differently from the quantum bosonic case (88a), where only quartic vertices are present. The interacting Lagrangian includes a three-legged (Fig. 10(a)) and a four-legged vertex (Fig. 10(b)), both featuring two incoming $\varphi$ fields which can be used to connect vertices to the two initial condition sources $n_0\tilde{\varphi}$. One then perturbatively expands both the cubic and quartic interaction vertices in $\Gamma$ and the initial boundary terms in $n_0$. The expansion of the initial boundary term is akin to the expansion we performed in Subsec. 4.3 in the simpler case of one-body decay (no interaction vertices therefore). For instance, the vertex $2\Gamma\tilde{\varphi}\varphi^2$ connects two sources to one surviving particle $\tilde{\varphi}$ via the Y-shaped diagram in Fig. 10(c), which is determined using Feynman rules. Clearly, higher-order diagrams may show a recursive structure, where fundamental graphs, e.g., the tree-level Y-shaped graphs, are nested with each other. Hence, considering a given vertex expanded to any order, one can write a Dyson equation for the *dressed* propagator, similarly to the quantum case in Eq. (94). The Dyson equation considers the infinitely many one-particle-reducible diagrams which can be drawn in terms of the considered vertex at the chosen order of the perturbative expansion. In our example, one can see that at first order in $\Gamma$, and second in $n_0$, the Y-shaped fundamental diagram of Fig. 10(a) can be recursively "mounted" into an infinite-order tree-level graph. One can then easily write a recursive equation for the dressed Green's function $G(\vec{x}, t)$, which in our picture corresponds to the homogeneous tree-level density $G(\vec{x}, t) = n_{\text{tree}}(t)$, leading to the phenomenological rate equation [8–10, 22, 23]:

$$\frac{dn_{\text{tree}}(t)}{dt} = -2\Gamma n_{\text{tree}}^2(t). \tag{A.4}$$

Hence, neglecting the four-fields vertex and considering in the Dyson equation only the vertex $-2\Gamma\tilde{\varphi}\varphi^2$ to tree-level, simply reproduces the mean-field result.

In the manuscript, we have discussed how Eq. (A.4) describes the dynamics in the reaction-limited regime $\Gamma/D \ll 1$, when the fast diffusive mixing erases spatial fluctuations in the density. For finite diffusive mixing $\Gamma/D \sim 1$, diffusion-limited regime, Eq. (A.4) describes the dynamics only in high spatial dimensions. In low dimensions, instead, spatial fluctuations are important and one has algebraic decay with exponent different from that predicted by mean field. This decay can be obtained by considering the one-loop correction to the tree-level density. Loop corrections are built by combining both interaction vertices, as depicted in the

diagram in Fig. 10(d). Sources $n_0$ must be connected to the right of the one-loop-level Dyson equation via two bare propagators as in Fig. 10(c). Internal momenta must now be integrated over the entire momentum space, due the internal loop, leading to $d$-dependent ultraviolet-divergent, for $d > 2$, and infrared-divergent, for $d < 2$, contributions. The latter are removed by systematic renormalisation. A detailed explanation of the renormalisation programme is detailed in Refs. [8–10, 22, 23]. We here report just the final result for the power-law decay for the sake of completeness:

$$\langle n \rangle(t) \sim \begin{cases} (Dt)^{-d/2}, & \text{if } d < 2, \\ \ln(Dt)/Dt, & \text{if } d = 2, \\ (\Gamma_{\text{eff}} t)^{-1}, & \text{if } d > 2, \end{cases} \tag{A.5}$$

with the effective reaction rate $\Gamma_{\text{eff}}$ defined by non-universal corrections stemmed from the loop expansion in $d > 2$. Crucially, a different decay exponent is identified depending on the space dimension being above or below the critical dimension $d_c = 2$. For $d = 1$, namely, decay $\langle n \rangle(t) \sim (Dt)^{-1/2}$ different from the mean field prediction (A.4) is observed. The diffusion-limited decay is controlled by the time two far apart particle take, on average, to meet and it is therefore determined by the diffusion constant $D$ (note, indeed, the different scaling of time compared to Eq. (A.4)). In $d = d_c = 2$ the mean field algebraic decay exponent is still valid but logarithmic corrections are present on top of it.

## B  Green's function in the presence of a slowly varying potential

In this Appendix, we report the derivation of the Green's function $G^{R,A}$ in Eq. (105) in the presence of an external potential $V(\vec{\bar{x}}) = V(\Gamma \vec{x})$. We assume the latter to be slowly varying on a length scale set set by $\Gamma$ according to (93). For the sake of convenience, we also set $\Gamma_d = 0$. The equation defining the Green's function is

$$\left\{ \delta(x_1 - x_2) \left[ i \partial_{t_2} + J \nabla^2_{x_2} - V \left( \Gamma \frac{\vec{x}_1 + \vec{x}_2}{2} \right) \right] \right\} \circ G_0^{R/A}(x_2, x_3) = \delta(x_1 - x_3). \tag{B.1}$$

The Wigner transform (43) of the previous equation yields

$$\left[ (\epsilon - Jk^2) + \frac{i}{2}(\partial_t + 2J\vec{k} \cdot \vec{\nabla}_x) + \frac{1}{4}J\nabla^2_x - V(\Gamma\vec{x}) e^{\frac{i}{2}(\overleftarrow{\partial_x}\overrightarrow{\partial_k} - \overleftarrow{\partial_k}\overrightarrow{\partial_x})} \right] G_0^{R/A}(x, k) = 1. \tag{B.2}$$

In order to simplify the previous equation, we now consider the Euler scaling limit (99). In this limit derivatives with respect to the slow centre of mass coordinate are small and therefore the expression (B.2) can be truncated at the first nonvanishing order in the derivatives in $x$. Accordingly, one retrieves the following simple expression for the retarded and advanced Green's functions in momentum space expressed in the rescaled coordinate $\bar{x} = \Gamma x$:

$$G_0^{R/A}(\bar{x}, k) = \frac{1}{\epsilon - Jk^2 - V(\vec{\bar{x}}) \pm i\delta}, \tag{B.3}$$

with $\delta > 0$ displacing the pole in the upper or lower complex half-plane, thus determining the $G_0^{R/A}(x, k)$ being retarded or advanced, respectively. Within the Euler-scaling limit, slow variations of the potential with respect to the centre of mass $x$ accordingly determine a slow dependence of the Green's functions on $x$. Clearly, the expression can be anti-Wigner transformed in the frequency variable. This leads to:

$$G_0^{R/A}(\vec{\bar{x}}, t, \vec{k}, t') = \mp i e^{-i\epsilon_k(\vec{\bar{x}})t'} \Theta(\pm t') = \mp i e^{-iJk^2 t' - iV(\vec{\bar{x}})t'} \Theta(\pm t'). \tag{B.4}$$

Therefore, the quasi-particle dispersion relation $\epsilon_k(\vec{\bar{x}}) = J\vec{k}^2 + V(\vec{\bar{x}})$ is *locally* modified by the presence of the external potential according to a local density approximation valid for a slowly varying potential. One can then anti-Wigner transform (B.4) with respect to the spatial momentum variable, in order to reach the final expression:

$$G_0^{R/A}(\vec{x}, t, \vec{x}', t') = \mp i \left[ \frac{i}{4\pi J t'} \right]^{d/2} \exp\left[ -i \frac{\vec{x}'^2}{4J t'} \right] e^{-iV(\vec{\bar{x}})t'} \Theta(\pm t'). \tag{B.5}$$

In the case of equal time arguments $t' = 0$, the previous expression yields the regularised retarded $G_{0,\varepsilon}^R$ and advanced $G_{0,-\varepsilon}^A$ Green's functions in Eq. (105) by putting $\Theta(\varepsilon) = 1$. We observe that the free Green's functions $G_0^{R/A/K}$ in case of a non-vanishing external potential $V$ are given by the product of two terms: the first term is translationally-invariant, solely depends on the relative coordinate $\vec{x}'$, and describes free quantum-ballistic propagation in a homogeneous space. Conversely, the second term is not translationally invariant as it encodes the confinement of quasiparticles modes due to the potential $V(\vec{x})$, and it depends on the centre-of-mass $\vec{\bar{x}}$ coordinate only. No explicit dependence on the centre of mass time variable $t$ appears, as time-translational invariance in the absence of boundary-initial terms still holds.

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
