# Peer review of "Kinetics of Quantum Reaction-Diffusion systems"

_SciPost Physics Core, doi:SciPost Phys. Core 8, 014 (2025)_

## Round 1 · Referee Report · Anonymous (Referee 1) · 2024-10-17

Report

I read the manuscript with great interest, and I believe that, while this submission does not meet the criteria for SciPost Physics, it would be more suited for SciPost Physics Lecture Notes, where it could potentially be published.

I thoroughly enjoyed reading this work, as it provides a comprehensive introduction and review of the results related to the Keldysh path integral formulation of reaction-diffusion systems. A topic that I consider important and timely.

My primary concern regarding the publication of this work (as also acknowledged by the authors several times) is that it primarily offers a more systematic presentation of the results from Ref. [56] with only minimal developments beyond them. However, I do believe that the paper succeeds in delivering a pedagogical and complete introduction to the results of Ref. [56], produced by the same authors. Therefore, I recommend its publication in SciPost Physics Lecture Notes.

Recommendation

Accept in alternative Journal (see Report)

---

## Round 1 · Referee Report · Anonymous (Referee 2) · 2024-10-28

Report

The manuscript provides a thorough and insightful analysis of open quantum systems with dissipative particle losses, addressing an important area at the intersection of quantum many-body physics and reaction-diffusion dynamics. By employing the Keldysh path-integral approach, the authors effectively generalize classical reaction-diffusion dynamics to the quantum domain.

The paper’s structure is clear and logical, guiding readers through the technicalities of Keldish path integral. The authors' derivation of a Boltzmann-like kinetic equation and analysis of particle density decay rates as a function of particle statistics represents a valuable contribution.

One particularly commendable aspect is the consideration of spatial inhomogeneity through the introduction of a trapping potential and the calculation of the dynamics of entanglement entropies.

Overall, this paper presents well-founded results that are likely to stimulate further theoretical and experimental work in open quantum dynamics.

I found the presentation and the clarity of the paper exemplary. Moreover, while some of the material was already presented in Ref. 56, the paper discusses new aspects, such as the entanglement entropies. Therefore I recommend publication is Scipost Phys. Concerning the entropies I think that the authors should compare their results with the framework developed in

Phys. Rev. B 105, 144305 (2022)
Phys. Rev. B 103, 020302 (2021)

and with

SciPost Phys. 12, 011 (2022)

where the case of localised sources of dissipation is
investigated.

Recommendation

Publish (meets expectations and criteria for this Journal)

---

## Round 1 · Referee Report · Anonymous (Referee 3) · 2024-12-1

Report

The authors signficantly expand the results presented in Ref. 56, the presentation is quite pedagogical and easy and enjoyable to follow. The questions are timely and the approach is quite detailed making it accessible to someone who is not an expert in Keldysh field theory. That said, I agree with another referee that the paper is more suited for another journal as the main results are already published in a different venue.

Recommendation

Accept in alternative Journal (see Report)

---

## Editorial Decision

published